



# Deglacial carbon cycle changes observed in a compilation of 117 benthic $\delta^{13}$C time series (20-6 ka)

Carlye Peterson[1,2] and Lorraine Lisiecki[2]

[1]Department of Earth Sciences, University of California Riverside, Riverside, California, USA.
[2]Department of Earth Science, University of California Santa Barbara, Santa Barbara, California, USA.
**Correspondence:** CARLYE PETERSON (CARLYE.PETERSON@GMAIL.COM)

**Abstract.** We present a compilation of 117 time series $\delta^{13}$C records from *Cibicides wuellerstorfi* spanning the last deglaciation (20-6 kyr) and well-suited for reconstructing large-scale carbon cycle changes, especially for comparison with isotope-enabled carbon cycle models. The age models for the $\delta^{13}$C records are derived from regional planktic radiocarbon compilations (Stern and Lisiecki, 2014). The $\delta^{13}$C records were stacked in nine different regions and then combined using volume-weighted
averages to create intermediate, deep, and global $\delta^{13}$C stacks. These benthic $\delta^{13}$C stacks are used to reconstruct mean changes in the size of the terrestrial biosphere and deep ocean carbon storage. The timing of change in global mean $\delta^{13}$C is interpreted to indicate terrestrial biosphere expansion from 19-6 ka. The $\delta^{13}$C gradient between the intermediate and deep ocean, which we interpret as a proxy for deep ocean carbon storage, matches the pattern of atmospheric $CO_2$ change observed in ice core records. The presence of signals associated with the terrestrial biosphere and atmospheric $CO_2$ indicates that the compiled
$\delta^{13}$C records have sufficient spatial coverage and time resolution to accurately reconstruct large-scale carbon cycle changes during the glacial termination.

## 1 Introduction

On glacial-interglacial timescales, carbon cycle changes redistribute the amount of carbon stored in the deep ocean, atmosphere
and terrestrial biosphere (*e.g.*, Broecker (1982); Siegenthaler et al. (2005)). For example, as atmospheric $CO_2$ increased across the deglaciation, atmospheric $\delta^{13}$C decreased, likely due to the ventilation of respired, $^{13}$C-depleted carbon from the deep ocean (*e.g.*, Schmitt et al. (2012); Eggleston et al. (2016)). However, identifying the biogeochemical mechanisms associated with these carbon transfers is complicated by a variety of carbon cycle feedbacks (*e.g.*, Archer et al. (2000); Sigman and Boyle (2000); Peacock et al. (2006); Toggweiler et al. (2006); Kohfeld and Ridgwell (2009); Brovkin et al. (2012); Menviel
et al. (2012); Galbraith and Jaccard (2015); Buchanan et al. (2016)). This study seeks to improve our understanding of glacial-interglacial carbon cycle changes by reconstructing changes in mean ocean $\delta^{13}$C and its vertical gradient and comparing the results with changes in the terrestrial biosphere and atmospheric $CO_2$ .



The $\delta^{13}$C of benthic foraminiferal calcite is a well-established carbon cycle proxy, which records the $\delta^{13}$C signature of the dissolved inorganic carbon (DIC) in seawater at seafloor depths (*e.g.*, Woodruff and Savin (1985); Zahn et al. (1986); Lutze and Thiel (1989); Duplessy et al. (1988); Mackensen (2008); Gottschalk et al. (2016); Schmittner et al. (2017)). Averages of benthic foraminiferal $\delta^{13}$C time series, called stacks, can improve the signal-to-noise ratio of regional or global seawater changes (*e.g.*,

Lisiecki et al. (2008); Lisiecki (2014)). Global mean benthic $\delta^{13}$C change is likely caused by changes in terrestrial organic carbon storage (Shackleton, 1977; Curry et al., 1988; Duplessy et al., 1988; Ciais et al., 2012; Peterson et al., 2014), while vertical $\delta^{13}$C gradients may record changes in deep ocean carbon storage and atmospheric $CO_2$ (Oppo and Fairbanks, 1990; Flower et al., 2000; Hodell et al., 2003; Lisiecki, 2010). The vertical $\delta^{13}$C gradient between the surface (high $\delta^{13}$C) and deep ocean (low $\delta^{13}$C) primarily results from the accumulation of low-$\delta^{13}$C respired organic carbon in deep water, which

temporarily sequesters it from the atmosphere. Conversely, vertical mixing of the ocean will tend to ventilate deep ocean carbon to the surface ocean and atmosphere while simultaneously decreasing the vertical $\delta^{13}$C gradient. Therefore, the vertical $\delta^{13}$C gradient likely records changes in deep ocean carbon storage, which is an important factor controlling glacial-interglacial changes in atmospheric $CO_2$ (*e.g.*, Schmitt et al. (2012); Eggleston et al. (2016)).

Here we compile and analyze 117 high-resolution benthic $\delta^{13}$C records from the Atlantic, Pacific, and Indian Oceans span-

ning the last deglaciation to investigate changes in both the ocean and terrestrial biosphere components of the global carbon cycle. Benthic $\delta^{13}$C records are combined into regional stacks, which are then used to construct time series of volume-weighted global mean $\delta^{13}$C and the vertical $\delta^{13}$C gradient between intermediate and deep waters.

We analyze these stacks to test the following hypotheses:

1. The deglacial pattern of global mean ocean $\delta^{13}$C change is a proxy for changes in the size of the terrestrial biosphere. If

so, global mean $\delta^{13}$C should continue to increase after atmospheric $CO_2$ levels plateau at 11 ka due to the slower response times for ice sheet retreat and ecosystem change (*e.g.*, Hoogakker et al. (2016); Davies-Barnard et al. (2017)). We compare the reconstructed global mean $\delta^{13}$C change with several carbon cycle model estimates of terrestrial biosphere change. Additionally, we evaluate whether deep Pacific $\delta^{13}$C correlates with global mean $\delta^{13}$C change as previously assumed (Shackleton et al., 1983; Curry and Oppo, 1997; Lisiecki et al., 2008). This study provides the first opportunity

to compare time series of deep Pacific $\delta^{13}$C with a volume-weighted global mean $\delta^{13}$C stack.

2. Changes in the vertical $\delta^{13}$C gradient should closely resemble time series of atmospheric $CO_2$ if the deglacial $CO_2$ increase is caused by a decrease in deep ocean carbon storage. This hypothesis is supported by findings on orbital timescales using a smaller number of sites (Oppo and Fairbanks, 1990; Flower et al., 2000; Hodell et al., 2003; Lisiecki, 2010), but the link between the vertical $\delta^{13}$C gradient and $CO_2$ has not yet been evaluated at millennial timescales

or using a global data compilation. Observing such a link would improve our understanding of deglacial atmospheric $CO_2$ increase and, furthermore, demonstrate that the data compilation presented here has adequate spatial and temporal resolution with sufficiently precise age models to reconstruct millennial-scale changes in global benthic $\delta^{13}$C.



## 2 BACKGROUND

### 2.1 Benthic $\delta^{13}$C reconstructions

Measurements of $\delta^{13}$C from the calcite tests of epibenthic foraminifera *Cibicides wuellerstorfi* and related species (Schweizer
et al., 2009) are commonly used to trace the spatial distribution of nutrients and deep water masses as well as changes in ocean

carbon cycling (*e.g.*, Curry et al. (1988); Duplessy et al. (1988); Curry and Oppo (2005); Schmittner et al. (2017)). However,
the *Cibicides* species *C. kullenbergi* and *C. mundulus*, often measured in deep South Atlantic cores, appear to record more
depleted $\delta^{13}$C values than *C. wuellerstorfi* (Gottschalk et al., 2016).

Mean $\delta^{13}$C change between the Last Glacial Maximum (LGM, 20 ka) and Late Holocene (6-0 ka) has been assessed with
global compilations of *Cibicides wuellerstorfi* $\delta^{13}$C records (*e.g.*, Shackleton (1977); Duplessy et al. (1988); Curry et al. (1988);

Boyle (1992); Matsumoto and Lynch-Stieglitz (1999); Curry and Oppo (2005); Herguera et al. (2010); Oliver et al. (2010);
Hesse et al. (2011); Peterson et al. (2014); Gebbie et al. (2015)). These time slice studies include as many as 500 core sites, but
generally undersample portions of the ocean with poor carbonate preservation, low primary productivity, and low sedimentation
rates (*i.e.*, the Southern Ocean south of 55S, the Indian Ocean, and the Pacific Ocean). In contrast, some portions of the Atlantic,
especially the North Atlantic, are relatively well-sampled with abundant, well-preserved *C. wuellerstorfi*. Therefore, the time

evolution of whole-ocean $\delta^{13}$C is less well-constrained than Atlantic vertical, zonal, and meridional $\delta^{13}$C gradients.

Because deglacial carbon cycle changes occur on millennial to centennial timescales, there is a need for a high resolution
benthic $\delta^{13}$C compilation of globally-distributed sites on a consistent age model across the glacial termination. Global compi-
lations of $\delta^{13}$C time series tend to focus on orbital-scale responses because their age models are not precise enough to analyze
the relative timing of carbon cycle changes during the deglaciation (*e.g.*, Lisiecki et al. (2008)). One global $\delta^{13}$C data synthe-

sis, which includes 258 records from many benthic and planktic foraminifera species, is not recommended for analyzing $\delta^{13}$C
changes on timescales of less than 10 kyr due to age model uncertainty and the inclusion of low-resolution records (Oliver
et al., 2010). Studies of the last glacial termination often focus on local or regional depth transects that contain high-resolution
$\delta^{13}$C records with good age control (*e.g.*, Sarnthein et al. (1994); Thornalley et al. (2010); Hoffman and Lund (2012); Tessin
and Lund (2013); Lund et al. (2015); Oppo et al. (2015); Sikes et al. (2016)). In modeling studies, transient simulations are typ-

ically compared to a small number of individual benthic $\delta^{13}$C records or regional syntheses, presumably due to the limitations
of available global $\delta^{13}$C compilations (*e.g.* Köhler et al. (2005); Brovkin et al. (2007); Köhler et al. (2010)).

### 2.2 Terrestrial biosphere and mean ocean $\delta^{13}$C

A portion of the additional carbon released from the deep ocean since the LGM was taken up by the terrestrial biosphere. The
transfer of carbon between the terrestrial biosphere and the deep ocean affects the global mean value of benthic $\delta^{13}$C because

the mean $\delta^{13}$C signature of the terrestrial biosphere is significantly more negative (approximately -25‰) than mean ocean
$\delta^{13}$C (approximately 0‰) (Shackleton, 1977). The change in global mean benthic $\delta^{13}$C between the LGM and the Holocene
is estimated to be 0.32‰ $\pm$ 0.20‰ (Peterson et al., 2014; Gebbie et al., 2015), but the timing of mean benthic $\delta^{13}$C change
across the deglaciation is not well known.



Deglacial changes in terrestrial carbon storage (soils and vegetation) can be reconstructed in many ways, including terrestrial vegetation proxies and archives (*e.g.*, pollen, paleovegetation), carbon cycle models (*e.g.*, box models, inverse methods, dynamic global vegetation models, biomization methods, *etc.*), and proxies such as benthic $\delta^{13}$C, triple oxygen isotopes (Landais et al., 2007), and atmospheric carbonyl sulfide (Aydin et al., 2016). These methods produce estimates of change in terrestrial

carbon storage between the LGM and Holocene that vary from 200-1900 PgC due to uncertainties and assumptions associated with each method (see discussion and citations within Peterson et al. (2014)).

Due to uncertainties in the total magnitude of change, here we focus on comparing the timing of changes in terrestrial carbon storage and global mean benthic $\delta^{13}$C. Models simulate rapid increases in terrestrial carbon storage from approximately 19-10 ka, followed by more gradual changes from 10-0 ka (Kaplan et al., 2002; Joos et al., 2004; Köhler et al., 2005). More recently,

the potential effects of changes in poorly-constrained carbon reservoirs (beneath ice sheets and on continental shelves) were evaluated using deglacial simulations of biogeophysical and land carbon changes from the HadCM3 General Circulation Model (GCM). The model simulated a rapid increase in terrestrial carbon storage from 20-14 ka, different responses between 14-11 ka depending on the model scenario, and then steady, gradual change from 11-4 ka (Davies-Barnard et al., 2017).

Estimates of global mean benthic $\delta^{13}$C are also used to remove global changes from individual $\delta^{13}$C records in order to

identify patterns of local or regional change, *e.g.*, related to ocean circulation. Because estimates of global mean $\delta^{13}$C have only been available for the LGM and Holocene, some studies use deep Pacific $\delta^{13}$C time series as a proxy for global mean $\delta^{13}$C change (Shackleton et al., 1983; Curry and Oppo, 1997; Lisiecki et al., 2008). Given the large volume and carbon storage capacity of the deep Pacific, its $\delta^{13}$C change should be similar in magnitude and timing to the mean ocean $\delta^{13}$C change; however, no study has yet confirmed this relationship. For example, low sedimentation rates and poor carbonate

preservation in the deep Pacific may limit how well deep Pacific $\delta^{13}$C time series resolve changes in mean ocean $\delta^{13}$C. Additionally, large changes in Atlantic or Indian Ocean $\delta^{13}$C could alter the timing of global mean $\delta^{13}$C relative to the Pacific. By constructing a global benthic $\delta^{13}$C stack, we can now directly compare deep Pacific $\delta^{13}$C with global mean $\delta^{13}$C change across the deglaciation.

## 2.3   Vertical gradients in benthic $\delta^{13}$C

A vertical gradient in the $\delta^{13}$C of DIC between surface/intermediate waters and deep water results from a combination of physical, chemical, and biological processes. The air-sea gas exchange of $CO_2$ between the atmosphere and surface ocean generates a temperature-dependent fractionation (Lynch-Stieglitz et al., 1995). Biological productivity in the surface ocean preferentially incorporates $^{12}$C into organic molecules, leaving $^{13}$C-enriched DIC in surface waters. Conversely, deep water becomes depleted in $^{13}$C due to remineralization of sinking organic carbon with a $\delta^{13}$C signature of approximately -25‰. The

accumulation of respired organic carbon in the deep ocean gradually increases deep water's DIC concentration while decreasing its $\delta^{13}$C value. Thus, sinking organic carbon simultaneously creates vertical gradients in both $\delta^{13}$C and DIC, creating low $\delta^{13}$C and high DIC in the deep ocean and high $\delta^{13}$C and low DIC in the surface ocean. However, deep water $\delta^{13}$C is also affected by the transport of relatively high-$\delta^{13}$C North Atlantic Deep Water into the deep Atlantic, where it mixes with low-$\delta^{13}$C waters from the Southern Ocean (Talley, 2013).



Numerous $\delta^{13}$C records from the well-characterized Atlantic Ocean demonstrate an enhanced vertical $\delta^{13}$C gradient between intermediate and deep water during the LGM (*e.g.*, Curry and Lohmann (1982); Curry et al. (1988); Duplessy et al. (1988); Sarnthein et al. (1994); Hodell et al. (2003); Curry and Oppo (2005); Marchitto and Broecker (2006); Herguera et al. (2010)). The less well-sampled Pacific and Indian Oceans also show signs of enhanced stratification at the LGM based on stronger

vertical $\delta^{13}$C gradients and other nutrient and ventilation proxies (*e.g.*, Kallel et al. (1988); Matsumoto and Lynch-Stieglitz (1999); Matsumoto et al. (2002); Herguera et al. (2010); Lund et al. (2011b); Allen et al. (2015); Sikes et al. (2016)).

Multiple causes have been proposed for stronger vertical $\delta^{13}$C gradients during the LGM, including increased surface productivity and export, increased ocean stratification, and changes in preformed $\delta^{13}$C in regions of deep water formation (*e.g.*, Matsumoto et al. (2002); Curry and Oppo (2005); Marchitto and Broecker (2006); Lynch-Stieglitz et al. (2007); Marinov et al.

(2008b, a); Herguera et al. (2010); Hesse et al. (2011); Lund et al. (2011a, b); Allen et al. (2015); Gebbie et al. (2015); Gloege et al. (2017); Menviel et al. (2017)). The large vertical $\delta^{13}$C gradient at the LGM could indicate a strong biological pump and/or weak vertical mixing, either of which would increase deep ocean carbon storage. Although studies do not agree about the relative importance of different mechanisms in creating this vertical gradient, the consensus is that the enhanced vertical $\delta^{13}$C gradient at the LGM is consistent with greater deep ocean carbon storage and that this carbon was transferred to the

atmosphere and terrestrial biosphere during the glacial termination.

On orbital timescales, changes in the intermediate-to-deep vertical $\delta^{13}$C gradient closely match atmospheric $CO_2$, with weaker vertical $\delta^{13}$C gradients corresponding to higher $CO_2$ levels (Oppo and Fairbanks, 1990; Flower et al., 2000; Hodell et al., 2003; Köhler et al., 2010; Lisiecki, 2010). This relationship supports the assertion that many of the processes affecting $CO_2$ also alter the vertical $\delta^{13}$C gradient. Model simulations suggest that multiple processes contribute to deglacial $pCO_2$

rise (Bauska et al., 2016), including ocean temperature increase, enhanced Southern Ocean mixing rates (and the role of sea ice) (*e.g.*, Franois et al. (1997); Crosta and Shemesh (2002); Gildor et al. (2002); Hodell et al. (2003); Paillard and Parrenin (2004)), decreased alkalinity and carbon inventories (Yu et al., 2014; Kerr et al., 2017), reduced biological pump (Buchanan et al., 2016), enhanced global ocean circulation (Buchanan et al., 2016), and coral reef growth (*e.g.*, Vecsei and Berger (2004)). Here we evaluate the relationship between atmospheric $CO_2$ and vertical $\delta^{13}$C change at millennial resolution across the

deglaciation. It is beyond the scope of this study to evaluate how much of the change in $CO_2$ and the vertical $\delta^{13}$C gradient at the LGM is associated with specific processes, such as changes in the biological pump (Archer et al., 2003; Köhler et al., 2005; Brovkin et al., 2007; Galbraith and Jaccard, 2015), deep water formation (McManus et al., 2004; Curry and Oppo, 2005) and/or Southern Ocean stratification (Lund et al., 2011b; Burke and Robinson, 2012).

## 3   Data

This study presents a compilation of 117 previously published benthic $\delta^{13}$C time series of *Cibicides wuellerstorfi* in per mil relative to Vienna PeeDee Belemnite (V.P.D.B.) (Figure 1; Table A1). Each record in the compilation spans the time range 20-6 ka. Analysis does not extend after 6 ka because cores from several data-sparse regions were either too low-resolution or missing sediment from 6-0 ka. We only include $\delta^{13}$C records with mean sample spacing better than 3 kyr (87% have a mean sample



spacing of less than 2 kyr). We excluded any records with sample gaps of 4 kyr or larger and excluded any cores affected by the phytodetritus effect ("Mackensen effect") as assessed by the original authors and the criteria from Peterson *et al.* (2014). We included one *C. kullenbergi* record from the deep South Atlantic (MD07-3076Q) (Waelbroeck et al., 2011) which may record a more negative $\delta^{13}$C value than *C. wuellerstorfi* at the LGM (Gottschalk et al., 2016). Additionally, we use some cores with

samples labeled "*C. spp*" that may include some *C. kullenbergi* (Table A1).

## 4 Methods

### 4.1 Stacking

Age models for cores were developed by aligning the benthic $\delta^{18}$O records to the regional stacks of *Stern and Lisiecki* (2014), which have age models based on planktic foraminiferal $^{14}$C ages. Because age model uncertainties are approximately 1-2

kyrs (Stern and Lisiecki, 2014), and some of the $\delta^{13}$C records analyzed have sample spacings of 2-3 kyr, our interpretation focuses on $\delta^{13}$C features with timescales of about 2 kyr or greater. For example, we do not expect to reconstruct abrupt changes associated with the onset of the Bölling-Alleröd or with centennial-scale $CO_2$ changes (Marcott et al., 2014). Because $\delta^{13}$C record resolution varies between sites, we interpolate the $\delta^{13}$C records to an even 1-kyr spacing, which introduces an additional source of uncertainty in the data. Although combining information from multiple records inherently risks distorting the true

ocean state, this risk is counterbalanced by the potential for improved signal-to-noise when estimating regional and global signals. In supplemental material, we provide the original, uninterpolated records from all 117 sites, which could be used for comparison with transient deglacial ocean circulation experiments.

We define regions within ocean basins based on the spatial patterns in benthic $\delta^{13}$C, for example, differing LGM $\delta^{13}$C values between intermediate and deep sites (Figure 2). In the North Atlantic, we separate the intermediate North Atlantic (INA,

0.5-2 km) from the upper deep North Atlantic (UDNA, 2-4 km) and the lower deep North Atlantic (LDNA, >4 km). Because the South Atlantic has fewer records than the North Atlantic (Table 1) and a different vertical $\delta^{13}$C structure (Figure 2), we define the intermediate South Atlantic (ISA) as 0.5-2.5 km and the deep South Atlantic (DSA) as >2.5 km. We separate the Indo-Pacific into four regions: the intermediate Indian (II, 0.5-2 km), intermediate Pacific (IP, 0.5-2 km), deep Indian (DI, >2 km), and deep Pacific (DP, (>2 km). The longitude boundaries between the Atlantic, Indian, and Pacific basins are the same as

in Peterson *et al.* (2014). To create regional stacks, we average the interpolated $\delta^{13}$C records in each region (Figure 2; Table A1). Most regions contain at least six cores; however, the intermediate and deep Indian regions each contain only two $\delta^{13}$C records.

To calculate an intermediate, deep, and global mean $\delta^{13}$C stacks (Figure 3, Figure 4), we calculated the volume of water in each region defined above (Table 1) and averaged the regional stacks using volume weighting as a percent of total volume

(using a depth range of 0.5-5 km). Thus, we represent regions proportional to their volume rather than over-representing well-sampled regions. These global stacks include benthic $\delta^{13}$C records from sites in the Atlantic, Indian, and Pacific Oceans but excludes the Southern Ocean, Arctic Ocean, and shallow inland seas. Additionally, this compilation only includes benthic $\delta^{13}$C records from below 0.5 km; therefore, we refrain from interpreting or making assumptions about $\delta^{13}$C above 0.5 km. Planktic



$\delta^{13}$C data suggest that mixed layer $\delta^{13}$C values may closely track atmospheric $\delta^{13}$C change (Eggleston et al., 2016; Hertzberg et al., 2016).

We calculate a vertical $\delta^{13}$C gradient ($\Delta\delta^{13}$C$_{I-D}$) using the volume-weighted intermediate and deep regional stacks from the Atlantic and Pacific. The Indian Ocean regional stacks are excluded from this vertical gradient calculation because each

Indian region includes only two cores, making these regional stacks more susceptible to noise. A global vertical $\delta^{13}$C gradient that includes the Atlantic, Indian, and Pacific Oceans (AIP $\Delta\delta^{13}$C$_{I-D}$) is provided in the supplemental materials (Figure A1, Table A2). Additionally, we construct a vertical $\delta^{13}$C gradient between the intermediate North Atlantic and deep Pacific ($\Delta\delta^{13}$C$_{(INA/2)-DP}$) (Lisiecki, 2010) and compare both representations of the vertical $\delta^{13}$C gradient to atmospheric $CO_2$.

We estimate stack uncertainty using Monte Carlo simulations that account for the effects of measurement uncertainty and

intra-region $\delta^{13}$C variability. Specifically, we generate 95% confidence intervals for the stacks using 10,000 bootstrapped iterations that randomly resample $\delta^{13}$C records from each region. During the resampling process, we also simulate measurement uncertainty in each record by adding Gaussian white noise with a standard deviation of 0.15‰ (Gebbie et al., 2015). Differences in the benthic $\delta^{13}$C stack between different Monte Carlo simulations, each with 10,000 iterations, is on the order of 0.02‰ at the LGM (20-19 ka) and even smaller for the Holocene (6 ka). It is beyond the scope of the current study to quantify

uncertainty associated with portions of the ocean for which there is no available data.

## 4.2 Comparison to CO$_2$

To compare the $\delta^{13}$C data to atmospheric $CO_2$ changes from 20-6 ka, we spliced together the atmospheric $CO_2$ records of Marcott et al. (2014) and Monnin et al. (2004). No correction was necessary to splice these records at 8.9860 ka because the $CO_2$ value of 265.45 p.p.m.v. at 8.9855 ka in Monnin et al. (2004) agrees well with the value of 265.2 p.p.m.v. at 8.9730 ka in

Marcott et al. (2014) (Figure 5). For quantitative comparison, we down-sampled the spliced $CO_2$ record by interpolating it to the same 1-kyr resolution of our benthic $\delta^{13}$C stacks.

Additionally, we examine the potential for differences in the timing of $CO_2$ and $\delta^{13}$C change that could be caused by lags in the climate system or age model uncertainty. We calculate correlation coefficients for different potential lags by interpolating the spliced $CO_2$ record with different time offsets, using lags that range from +1000 yr to -1000 yr in increments of 100 yr.

For example, a 100-yr lag in $CO_2$ relative to the vertical $\delta^{13}$C gradient would be represented by comparing $\delta^{13}$C values at 6, 7, ... 20 ka with $CO_2$ values at 5.9, 6.9, ..., and 19.9 ka. Conversely, a $CO_2$ lead of 100 yr would be suggested if the correlation between the two is maximized for $CO_2$ values at 6.1, 7.1, ..., 20.1 ka.

Testing the significance of correlations between $\delta^{13}$C and $CO_2$ is complicated by the fact that both time series are autocorrelated, *i.e.*, each data point is highly correlated with the value immediately before or after. To reduce the impact of

autocorrelation, we pre-whiten the data by taking the difference between successive 1-kyr samples before calculating the linear correlation and its statistical significance. Our assessment of the statistical significance of the correlations accounts for the reduction in the number of degrees of freedom in the data associated with pre-whitening and/or allowing time lags between $\delta^{13}$C and $CO_2$ observations.



## 5   Results

### 5.1   Comparison to LGM and Holocene reconstructions

Although our time series compilation has fewer core sites than some previous studies of LGM $\delta^{13}$C, it preserves the large-scale features of these LGM $\delta^{13}$C reconstructions, such as enhanced vertical and meridional Atlantic $\delta^{13}$C gradients (Figure 2) (*e.g.*,
Curry and Oppo (2005); Peterson et al. (2014)). Vertical $\delta^{13}$C gradients are strongest in the glacial North Atlantic, closely followed by the glacial South Atlantic (Peterson et al., 2014). Indo-Pacific $\delta^{13}$C values are more depleted than North Atlantic $\delta^{13}$C values of the same depth. The most depleted $\delta^{13}$C values in the compilation are from the high-latitude deep South Atlantic during the LGM, possibly due to inclusion of data from *C. kullenbergi* (Gottschalk et al., 2016). Equatorial deep South Atlantic records at the LGM have $\delta^{13}$C values similar to equatorial deep Indo-Pacific $\delta^{13}$C records. However, our compilation lacks
Indo-Pacific sites deeper than 3.5 km.

At 6 ka the $\delta^{13}$C values in this compilation generally resemble the Holocene compilation of Peterson et al. (2014). Minor differences could result from Peterson *et al.* (2014) using Holocene data from 6-0 ka and including more North Pacific sites and more sites from 0.5-1.5 km depth.

### 5.2   Regional stacks

We create nine regional $\delta^{13}$C stacks from 20-6 ka (Figure 3, Table 1). Six of the regional $\delta^{13}$C stacks increase steadily from approximately 20-18 ka to 6 ka (LDNA, DSA, II, IP, DI, DP). Small deviations in the trend of the Indian $\delta^{13}$C stacks are interpreted as noise because these stacks each contain only two cores. Three Atlantic regions (INA, ISA, UDNA) show a decrease in $\delta^{13}$C from approximately 19-15 ka and then an increase from 14-6 ka, as described in previous studies (*e.g.*, Hodell et al. (2008); Thornalley et al. (2010); Hodell et al. (2010); Lund et al. (2011a); Tessin and Lund (2013); Oppo et al. (2015)).
The UDNA $\delta^{13}$C stack is briefly similar to the LDNA at 16 ka and then resembles the ISA stack from 14-6 ka. The DI and DSA $\delta^{13}$C values are generally similar across the deglaciation except that the DSA $\delta^{13}$C begins increasing at 18 ka while the DI $\delta^{13}$C increase begins at 16 ka. The intermediate-depth $\delta^{13}$C stacks in the Indian and Pacific Oceans are very similar for most of the time interval.

Across the deglaciation, the vertical $\delta^{13}$C gradient weakens in the Atlantic, most noticeably in the North Atlantic (INA-
LDNA: 1.20‰ at 20 ka and 0.31‰ at 6 ka). Vertical gradients in the Indian and Pacific Oceans show much less change. The largest spread in $\delta^{13}$C values is observed from 20-18 ka, when the intermediate North Atlantic and deep South Atlantic regions differ by 1.6‰; this gradient has decreases to 0.47‰ at 11-10 ka. The maximum difference between regions at 6 ka is 0.91‰ between the intermediate North Atlantic (most enriched) and the deep Indian (most depleted).

### 5.3   Volume-weighted stacks and global mean $\delta^{13}$C stack

A global mean $\delta^{13}$C stack is constructed by volume weighting all nine regional stacks. However, we construct two different versions of the intermediate and deep $\delta^{13}$C stacks, with and without the Indian stacks, because the Indian regions each contain



only two records. Both versions of the intermediate and deep stacks show similar trends, but we focus our analysis on the version that uses only the Atlantic and Pacific regions, which should be less susceptible to noise (Figure 4, Table 1). Results for the intermediate and deep $\delta^{13}$C stacks that include Indian Ocean are provided in supplemental materials (Figure A1, Table A2). The volume-weighted intermediate, deep, and global mean $\delta^{13}$C stacks increase across the deglaciation, but the magnitude of change is larger for the deep $\delta^{13}$C stack (0.46‰) than the intermediate $\delta^{13}$C stack (0.24‰) (Table 1, Figure 4). We define the vertical $\delta^{13}$C gradient, $\Delta\delta^{13}C_{I-D}$, as the difference between the volume-weighted Atlantic and Pacific intermediate and deep stacks. This gradient is largest at 20 ka (0.40‰) and shrinks to 0.24‰ by 6 ka.

The volume-weighted global $\delta^{13}$C stack holds nearly steady from 20 to 19 ka at approximately 0.00‰ (95% CI: -0.10 to 0.09‰ at 19 ka) and then increases from 18-6 ka, reaching a value of 0.37‰ (95% CI: 0.26 to 0.49‰) at 6 ka. The change from 20 to 6 ka in the global stack is 0.36‰ (95% CI: 0.25 to 0.46‰), which agrees to within uncertainty with the LGM-to-Holocene mean $\delta^{13}$C change estimate of 0.38‰ (95% CI: 0.30 to 0.46‰) for 0.5-5 km from (Peterson et al., 2014). Recall that the mean $\delta^{13}$C estimate from our global stack is not quite a whole-ocean $\delta^{13}$C estimate because we do not include data from the surface (<0.5 km), Southern Ocean (>65S), or bottom waters (>5 km). Estimates of whole-ocean $\delta^{13}$C change are slightly smaller at 0.34‰ (95% CI: 0.15 to 0.53‰) (Peterson et al., 2014) and 0.32‰ (95% CI: 0.12 to 0.52‰) (Gebbie et al., 2015). The surface ocean (0-0.5 km) has enriched $\delta^{13}$C values with much less deglacial change (Eggleston et al., 2016; Hertzberg et al., 2016).

## 6 Discussion

### 6.1 Terrestrial carbon storage and global mean benthic $\delta^{13}$C

The long-standing explanation for mean benthic $\delta^{13}$C change across the deglaciation is an increase in the size of the terrestrial biosphere (Shackleton, 1977; Curry et al., 1988; Duplessy et al., 1988). Here we compare the timing of changes in our global mean $\delta^{13}$C stack (*i.e.*, a monotonic increase from 19-6 ka, Figure 4B) with model simulations and other terrestrial biosphere reconstructions.

A carbon isotope-enabled transient model from LPJ-DVGM simulated a mean ocean $\delta^{13}$C increase beginning at 21 ka, with the most rapid changes occurring from 17-10.5 ka (Kaplan et al., 2002). In these experiments, the terrestrial biosphere began expanding around 18-16.5 ka (Joos et al., 2004; Köhler et al., 2005) and rapidly increased from 17-9 ka, with 70% of terrestrial carbon storage change occurring before the Holocene (11.5 ka) (Kaplan et al., 2002). In our global $\delta^{13}$C stack, 67% of the $\delta^{13}$C change occurs between 19-11 ka while the remaining 33% of $\delta^{13}$C change occurs from 11-6 ka.

Simulations from HadCM3 estimated that 45-70% of terrestrial biosphere expansion occurred between 18-14 ka (Davies-Barnard et al., 2017). Dramatically different trends were observed from 14-6 ka in simulations with different assumptions about carbon storage under glacial ice sheets and on continental shelves. The simulation that most closely resembles our global mean $\delta^{13}$C stack is the simulation that releases carbon from under ice sheets to the atmosphere and does not accumulate carbon on exposed continental shelves (Figure 5). This simulation is also the only one which agrees with terrestrial carbon storage change estimates of 440 PgC based on whole-ocean mean $\delta^{13}$C change (*e.g.*, Peterson et al., 2014).



Climate model simulations of the Holocene using a global pollen synthesis, the biomization method, and vegetation models (HadCM3, FAMOUS, and BIOME4) suggest that the global average area for most carbon-rich megabiomes (i.e., excluding grasslands and dry shrubland) increased from 10-2 ka and net primary productivity increased from 8-2 ka (Hoogakker et al., 2016). This is consistent with our observation that the global mean benthic $\delta^{13}$C trend continued until at least 6 ka. Dramatic

land use changes from agricultural practices, another potential mechanism for terrestrial carbon change, did not begin until 4.5 ka (Ruddiman and Ellis, 2009). More detailed evaluation of Holocene terrestrial carbon storage changes will require improved spatial coverage for $\delta^{13}$C records from 6-0 ka.

## 6.2 Global mean $\delta^{13}$C and deep Pacific $\delta^{13}$C

Previous studies have assumed deep Pacific $\delta^{13}$C can be used as a proxy for global mean $\delta^{13}$C because the deep Pacific

constitutes about 30% of the ocean volume and is not strongly affected by shifting water mass boundaries (*e.g.*, Shackleton et al. (1983); Curry and Oppo (1997); Lisiecki et al. (2008)). From 20-6 ka, the global mean and deep Pacific $\delta^{13}$C stacks show similar patterns of change (Figure 6) and fall along a tight regression line

$$\delta^{13}C_{global} = 1.02 \pm 0.06‰ \times \delta^{13}C_{DP} + 0.18 \pm 0.01‰$$

The two time series are highly correlated ($r^2$=0.99), which is not surprising because the large volume of the deep Pacific exerts

a strong influence on the global mean $\delta^{13}$C stack. However, when the stacks are pre-whitened to account for autocorrelation, their correlation falls on the edge of statistical significance (p=0.05). The statistical significance of this correlation is likely limited by the relatively short time interval analyzed and/or noise in the deep Pacific stack, which contains only seven $\delta^{13}$C records.

A carbon cycle box model simulated a strong correlation between deep Pacific $\delta^{13}$C and $CO_2$ across several glacial cycles

($r^2$=0.96) (Köhler et al., 2010). The correlation between our deep Pacific $\delta^{13}$C stack and $CO_2$ is statistically significant after pre-whitening ($r^2$=0.62, p=0.01), but global mean $\delta^{13}$C and $CO_2$ are not ($r^2$=0.20, p=0.24). Our compilation of Pacific records is likely insufficient to determine whether deep Pacific $\delta^{13}$C correlates better with global mean $\delta^{13}$C or atmospheric $CO_2$. This issue could be better resolved using a $\delta^{13}$C compilation spanning multiple glacial cycles and including more deep Pacific sites.

## 6.3 Vertical $\delta^{13}$C gradient and atmospheric $CO_2$

The vertical $\delta^{13}$C gradient ($\Delta\delta^{13}C_{I-D}$) in our compilation resembles the inverse of $CO_2$ change across the deglaciation (Figure 7), as would be expected if they are both strongly influenced by changes in deep ocean carbon storage (Flower et al., 2000; Oppo and Horowitz, 2000; Hodell et al., 2003). Alternatively, one orbital-scale study found a stronger correlation with $CO_2$ using the gradient between the deep Pacific and half the INA $\delta^{13}$C stack ($\Delta\delta^{13}C_{(INA/2)-DP}$) Lisiecki (2010). Both vertical $\delta^{13}$C gradients ($\Delta\delta^{13}C_{I-D}$ and $\Delta\delta^{13}C_{(INA/2)-DP}$) decrease from 18-11 ka over the same time interval that atmospheric

$CO_2$ increases. In contrast, the global mean $\delta^{13}$C stack increases at a relatively steady pace from 19-6 ka. Thus, the vertical $\delta^{13}$C gradient records a distinctly different signal than the global mean $\delta^{13}$C.

The vertical $\delta^{13}$C gradients decrease most rapidly across two time steps, 18-17 ka and 12-11 ka. The first change at 18 ka is approximately synchronous with the start of atmospheric $CO_2$ rise (Marcott et al., 2014) and a decrease of 0.3‰ in the $\delta^{13}$C



of atmospheric $CO_2$ (Eggleston et al., 2016). In the Southern Ocean at 18 ka, proxy records indicate a decrease in aeolian dust deposition accompanied by lower marine productivity (Martínez-Garcia et al., 2009) and a decrease in winter sea ice cover, which likely reduced vertical stratification (Ferrari et al., 2014). The second rapid change in the vertical $\delta^{13}C$ gradients at 12 ka approximately coincides with rapidly increasing atmospheric $CO_2$ from 13-11.5 ka and a decrease of 0.1‰ in the $\delta^{13}C$ of

atmospheric $CO_2$.

From 11 to 6 ka, atmospheric $CO_2$ remains nearly constant with a small (approximately 10 ppm) decrease from 11-8 ka. The vertical $\delta^{13}C$ gradients are also relatively steady from 11-6 ka, with a slight increase in both gradients from 9-8 ka. The small decrease in atmospheric $CO_2$ beginning at 11 ka (Marcott et al., 2014) has been variously attributed to growth of the terrestrial biosphere, sea level rise, and an increase in gas exchange through reduced sea ice cover (Kaplan et al., 2002; Joos et al., 2004;

Köhler and Fischer, 2004; Köhler et al., 2005, 2010).

Although $CO_2$ correlates strongly with both $\Delta\delta^{13}C_{I-D}$ ($r^2$=-0.98) and $\Delta\delta^{13}C_{(INA/2)-DP}$ ($r^2$=-0.98), we must pre-whiten these time series to remove autocorrelation before assessing the statistical significance of their correlation. At the 95% confidence level, atmospheric $CO_2$ significantly correlates with both $\Delta\delta^{13}C_{I-D}$ ($r^2$=-0.70; p<0.01) and $\Delta\delta^{13}C_{(INA/2)-DP}$ ($r^2$=-0.79; p<0.01) (Figure A1, Table 2). Better correlation with $\Delta\delta^{13}C_{(INA/2)-DP}$ could be because of better age con-

trol and higher resolution $\delta^{13}C$ records in the INA region than the other intermediate regions. Determining whether the $\Delta\delta^{13}C_{(INA/2)-DP}$ gradient or the volume-weighted vertical $\delta^{13}C$ gradient correlates better with atmospheric $CO_2$ will require data with better spatial coverage and/or a longer time span.

Because our comparison of the vertical $\delta^{13}C$ gradient and $CO_2$ could be affected by differences in timing caused by carbon cycle processes or age model uncertainty, we additionally investigate whether the correlations between $CO_2$ and the vertical

$\delta^{13}C$ gradient would be improved by age model shifts (Table 2). The correlation between $CO_2$ is maximized when $\Delta\delta^{13}C_{I-D}$ lags $CO_2$ by 300 years or when $\Delta\delta^{13}C_{(INA/2)-DP}$ leads $CO_2$ by 200 years (Table 2), both of which are within the age uncertainty of the sediment core age models. Thus, changes in atmospheric $CO_2$ and vertical $\delta^{13}C$ gradients appear synchronous to within age model uncertainty.

The prevailing processes that potentially explain atmospheric $CO_2$ change during glacial cycles include the efficiency of the

biological pump (Martínez-Garcia et al., 2009; Galbraith and Jaccard, 2015), circulation changes (Ferrari et al., 2014; Lacerra et al., 2017; Menviel et al., 2017; Sikes et al., 2017; Wagner and Hendy, 2017), or a combination of these and/or other processes (Bauska et al., 2016; Skinner et al., 2017). These processes could influence the carbon cycle on different timescales (Bauska et al., 2016; Kohfeld and Chase, 2017) and/or in different regions (*e.g.*, (Gu et al., 2017)), which could confound interpretations of which processes are responsible for atmospheric $CO_2$ change. However, because both productivity and circulation change

would affect the vertical $\delta^{13}C$ gradient while changing atmospheric $CO_2$, we interpret our results as supporting the importance of the deep ocean as a reservoir for storing glacial carbon from either or both processes. Furthermore, these results support the use of vertical $\delta^{13}C$ gradients as a proxy for glacial-interglacial $CO_2$ change (Lisiecki, 2010).




## 7 Conclusions

We present regional $\delta^{13}$C stacks and volume-weighted intermediate, deep, and global mean $\delta^{13}$C stack from a compilation of 117 benthic *C. wuellerstorfi* $\delta^{13}$C records, which span 20 to 6 kyr with a mean age resolution better than 2 kyr. Age models are based on $\delta^{18}$O alignments to regional stacks with radiocarbon dating, with age model uncertainties for each core

of approximately 1-2 kyr. The volume-weighted mean $\delta^{13}$C change estimate of 0.36‰ (95% CI: 0.25 to 0.46‰) from the 117 cores in this study is similar to the estimate of Peterson *et al.* (2014) based on 480 cores. This compilation also shows spatial patterns of benthic $\delta^{13}$C that are similar to higher resolution reconstructions of the Holocene and Last Glacial Maximum time slices.

The global mean $\delta^{13}$C stack is interpreted as recording an increase in the size of the terrestrial biosphere from 19 to 6 ka,

in agreement with modeling studies (Kaplan et al., 2002; Joos et al., 2004). To constrain the timing of the end of terrestrial biosphere expansion, future work should focus on extending this estimate through the Late Holocene. Furthermore, the $\delta^{13}$C changes from 20 ka to 6 ka suggest that a deep Pacific $\delta^{13}$C stack can be used as a proxy for global mean $\delta^{13}$C with an offset of 0.18‰. Future work should aim to validate this result over one or more glacial cycles and using more deep Pacific records. Both vertical $\delta^{13}$C gradients, between intermediate and deep water ($\Delta\delta^{13}C_{I-D}$) and intermediate North Atlantic

and deep Pacific ($\Delta\delta^{13}C_{(INA/2)-DP}$), are interpreted as proxies for changes in deep ocean carbon storage. We find that millennial-scale features in $\Delta\delta^{13}C_{I-D}$ and $\Delta\delta^{13}C_{(INA/2)-DP}$ are significantly correlated with atmospheric $CO_2$ changes from 20-6 ka. Based on these comparisons, we conclude that the four-dimensional compilation of globally distributed $\delta^{13}$C time series presented here provides useful constraints for global carbon cycle reconstructions and for comparison with deglacial simulations from isotope-enabled Earth systems models.

*Data availability.* The original data and publication citations along with this data synthesis are made available in supplemental files.



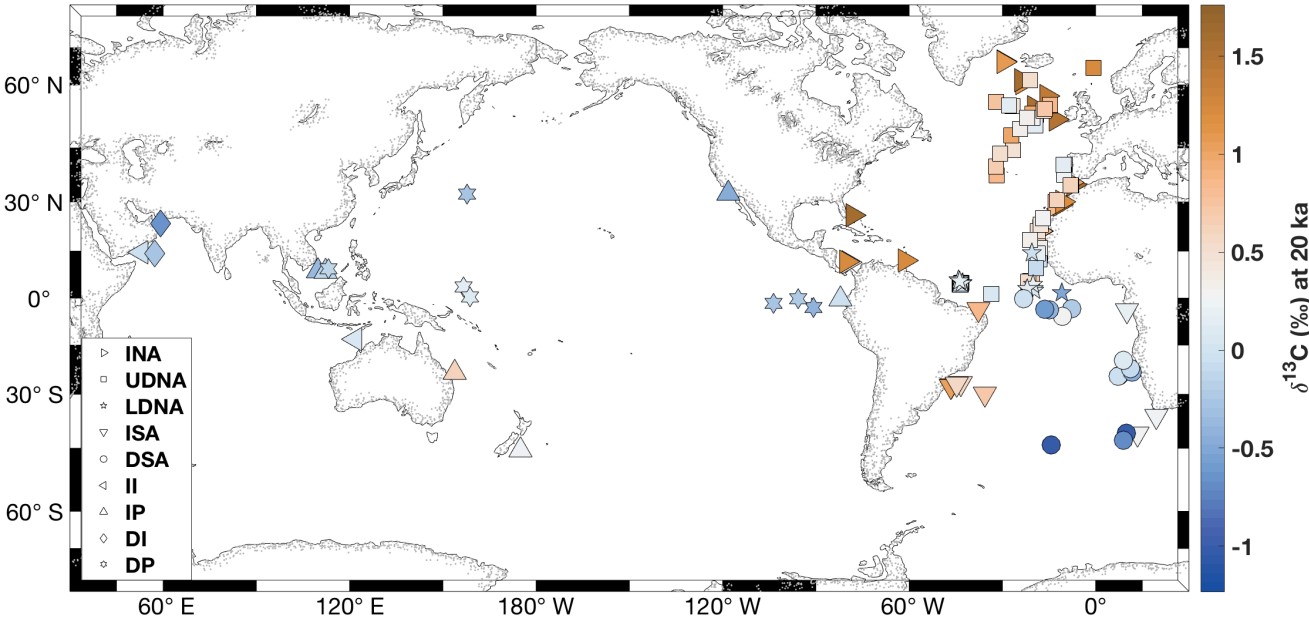

**Figure 1.** Locations of 117 core sites compiled for this study, color coded by LGM $\delta^{13}C$ estimates at each core site. Markers indicate locations of cores in the nine regions: INA = intermediate North Atlantic; UDNA = upper deep North Atlantic; LDNA = lower deep North Atlantic; ISA = intermediate South Atlantic; DSA = deep South Atlantic; II = intermediate Indian; DI = deep Indian; IP = intermediate Pacific; DP = deep Pacific.



**Figure 2.** The three-dimensional structure of $\delta^{13}$C in Indian and Pacific Oceans (a and c) and Atlantic (b and d) shown as zonally collapsed cross sections (latitude *vs.* depth) with the same marker scheme as in Figure 1. Dotted lines indicate region boundaries. Colors show the $\delta^{13}$C value at each site for the LGM (20 ka, top row) and Holocene (6 ka, bottom row). Note that latitudes on the x-axis are oriented so the Southern Ocean is in the center of the figure. Additional time slices (in 1-kyr intervals from 20-6 ka) and an animation of deglacial $\delta^{13}$C changes can be found in the supplemental materials.





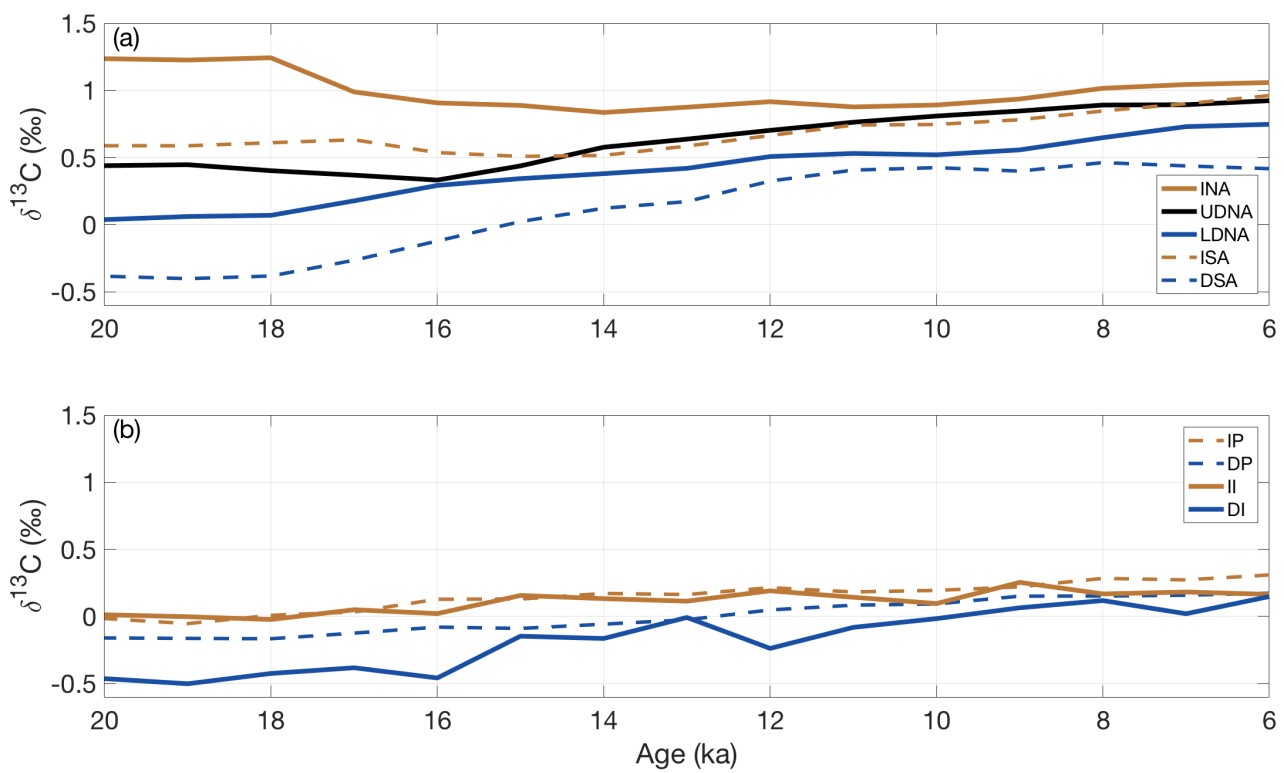

**Figure 3.** Regional stacks for the (a) Atlantic and (b) Indian and Pacific Oceans. Note the x- and y-axes are identically scaled. INA = intermediate North Atlantic; UDNA = upper deep North Atlantic; LDNA = lower deep North Atlantic; ISA = intermediate South Atlantic; DSA = deep South Atlantic; II = intermediate Indian; DI = deep Indian; IP = intermediate Pacific; DP = deep Pacific.





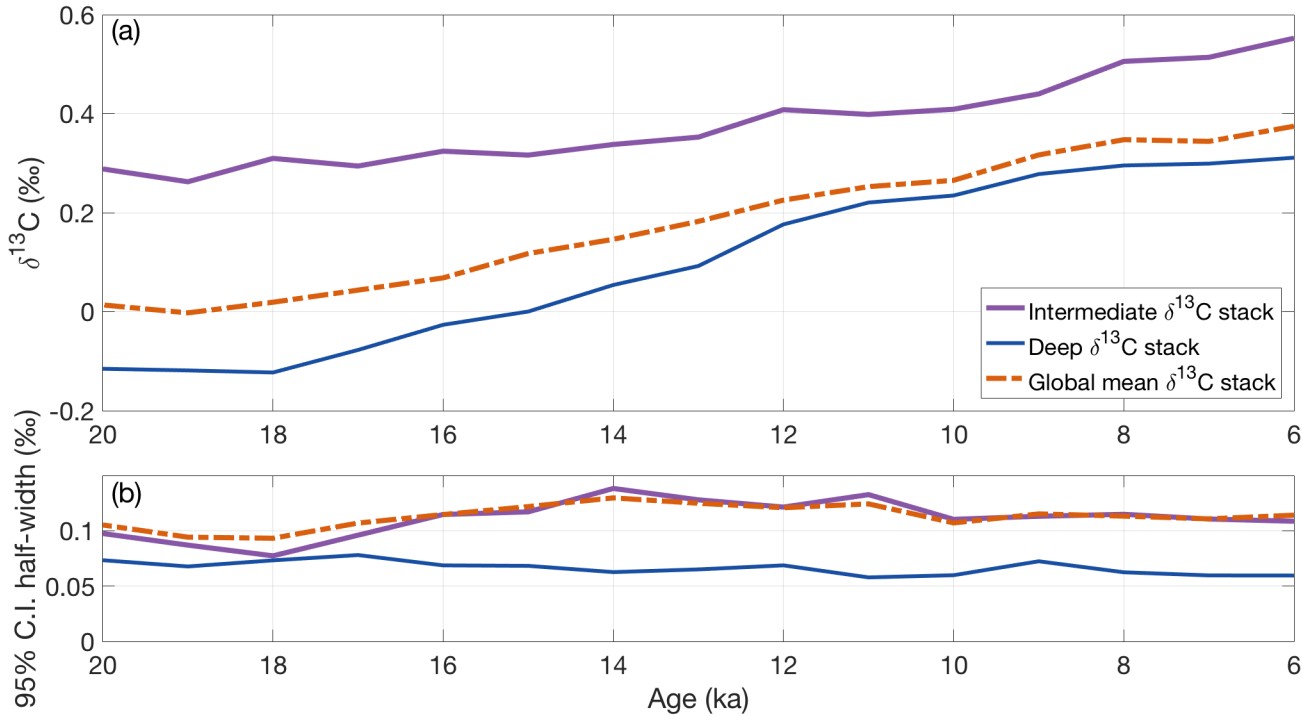

**Figure 4.** Volume-weighted stacks. (a) The volume-weighted global stack is calculated based on all nine regional stacks. However, the intermediate and deep stacks shown here only include Atlantic and Pacific data due to the small amount of Indian data. A comparison of these stacks with ones which include the Indian stacks is provided in Figure A1. (b) 95% confidence interval half-width showing the change in uncertainty across the deglaciation for each stack.





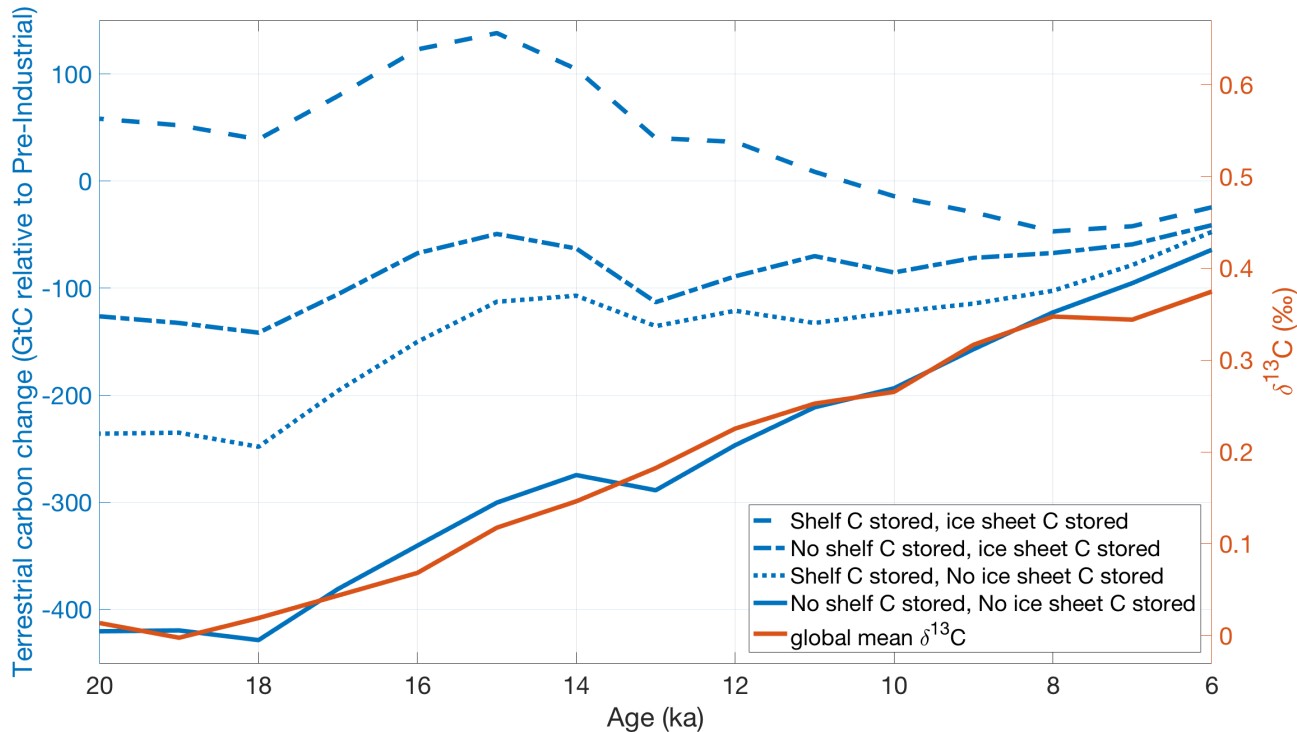

**Figure 5.** Time series of various HadCM3 simulations of terrestrial carbon storage (biosphere and soils) anomalies relative to the pre-Industrial plotted in blue (Davies-Barnard et al., 2017) and our global mean $\delta^{13}$C approximation of terrestrial carbon storage plotted in orange. Global mean $\delta^{13}$C change closely resembles one simulation (solid blue) that releases carbon from under ice sheets to the atmosphere and does not store carbon on continental shelves.





**Figure 6.** (a) Time series of global mean ocean $\delta^{13}C$ stack and deep Pacific $\delta^{13}C$ stacks. (b) Half-width 95% confidence intervals for the global mean stack and deep Pacific stack. (c) Deep Pacific $\delta^{13}C$ stack *vs.* global mean $\delta^{13}C$ stack. Each point is the $\delta^{13}C$ value for one time slice with 95% confidence intervals (vertical and horizontal error bars). Time across the deglaciation progresses toward the upper right corner. The best-fit line for the two stacks is plotted as a solid line with 95% confidence interval dashed lines.






**Figure 7.** Comparison of atmospheric $CO_2$ with both AP vertical $\delta^{13}$C gradient (a) and $\Delta\delta^{13}C_{(INA/2)-DP}$ gradient (b). Both vertical gradients closely resemble the spliced atmospheric $CO_2$ records (red circles) (Marcott et al., 2014; Monnin et al., 2004), but the $\Delta\delta^{13}C_{(INA/2)-DP}$ gradient is better correlated with atmospheric $CO_2$ from 20-6 ka. Recall the $\Delta\delta^{13}C_{(INA/2)-DP}$ gradient is not volume-weighted while the vertical AP $\delta^{13}$C gradient is volume-weighted, hence the right y-axes between the top and bottom panels are not directly comparable.





**Table 1.** Region details: number of sites included, the regional volume as a percent of the core depth range (0.5-5 km, excluding shallow inland seas, Southern Ocean, and Arctic Ocean), and the mean $\delta^{13}$C estimates at the LGM (20 ka), Holocene (6 ka), and the Holocene $\delta^{13}$C minus LGM $\delta^{13}$C difference. In parenthesis are bootstrapped mean $\delta^{13}$C values and 95% confidence interval for the global mean $\delta^{13}$C.

| Region name | Sites in stack | %Volume* | $\delta^{13}C_{Hol}$(‰) | $\delta^{13}C_{LGM}$(‰) | $\Delta\delta^{13}C_{Hol-LGM}$(‰) |
|---|---|---|---|---|---|
| INA | 18 | 5.0 | 1.06 | 1.24 | -0.18 |
| ISA | 9 | 7.9 | 0.96 | 0.59 | 0.37 |
| II | 2 | 8.0 | 0.16 | 0.01 | 0.15 |
| IP | 6 | 24.3 | 0.31 | -0.02 | 0.32 |
| Intermediate$_{w,m}$ | 35 | 45.2 | 0.55 | 0.29 | 0.26 |
| UDNA | 49 | 5.4 | 0.92 | 0.44 | 0.48 |
| LDNA | 10 | 1.5 | 0.75 | 0.04 | 0.71 |
| DSA | 14 | 6.2 | 0.42 | -0.38 | 0.80 |
| DI | 2 | 9.5 | 0.15 | -0.46 | 0.61 |
| DP | 7 | 32.2 | 0.17 | -0.16 | 0.33 |
| Deep$_{w,m}$ | 82 | 54.8 | 0.31 | -0.12 | 0.43 |
| Global$_{w,+}$ | 117 | 77.7* | 0.37 | 0.01 | 0.36 (95% CI: 0.25 to 0.46) |

*Volume of core depth range (0.5-5 km) as a proportion of global volume

$_w$Volume-weighted $\delta^{13}$C values

$_m$Excluding the Indian Ocean regions

$_+$Atlantic, Indian, and Pacific Ocean regions



**Table 2.** Correlation coefficients and p-values between pre-whitened records. We pre-whitened to account for autocorrelated time series and calculated p-values to account for reduced degrees of freedom for pre-whitened and/or time lagged correlations. To investigate possible leads/lags between records, we shift the atmospheric $CO_2$ record in 100-year increments relative to the $\delta^{13}C$ stacks. The atmospheric $CO_2$ record is spliced from $CO_2$ records from Monnin et al. (2004) at 8.9855 ka and Marcott et al. (2014) at 8.9730 ka.

| Record 1 | Record 2 | $CO_2$ time shift (years) | Pre-whitened $r^2$ | Pre-whitened p-value |
|---|---|---|---|---|
| Global mean $\delta^{13}C$ stack | Deep Pacific $\delta^{13}C$ stack | 0 | 0.46 | 0.05 |
| $CO_2$ | Deep Pacific $\delta^{13}C$ stack | 0 | 0.62 | 0.008 |
| $CO_2$ | Global $\delta^{13}C$ stack | 0 | 0.20 | 0.24 |
| $CO_2$ | Global $\delta^{13}C$ stack | $-500$ | 0.35 | 0.12 |
| $CO_2$ | $\Delta\delta^{13}C_{I-D}$ | 0 | -0.70 | 0.002 |
| $CO_2$ | $\Delta\delta^{13}C_{I-D}$ | $+300$ | -0.73 | 0.002 |
| $CO_2$ | $\Delta\delta^{13}C_{(INA/2)-DP}$ | 0 | -0.79 | 0.0002 |
| $CO_2$ | $\Delta\delta^{13}C_{(INA/2)-DP}$ | $-200$ | -0.82 | 0.0002 |



## Appendix A

**Figure A1.** (a) Two versions of the deep and intermediate stacks (colors in legend) with the AIP stacks plotted in dot-dashed lines, and AP stacks plotted in solid lines. (b) The 95% C.I. half-width for each stack in the legend in figure (A1a). (c) Comparison of $CO_2$ and both AP and AIP vertical $\Delta\delta^{13}C$ gradients. Both volume-weighted vertical gradients closely resemble the spliced atmospheric $CO_2$ records (red circles) (Marcott et al., 2014; Monnin et al., 2004), but the AIP $\Delta\delta^{13}C$ gradient (dashed green-blue line) is more noisy and slightly more depleted than the AP $\Delta\delta^{13}C$ gradient (solid blue line).



**Table A1.** Supplemental table of the name, location, region, and reference for each record in this $\delta^{13}$C synthesis

| Core Name | Lat | Lon | Depth (m) | Region | Reference |
|---|---|---|---|---|---|
| EW9209-1JPC | 5.9 | -44.2 | 4056 | LDNA | Curry and Oppo (1997) |
| GeoB7920-2 | 20.8 | -18.6 | 2278 | UDNA | Tjallingii et al. (2008) |
| GeoB9508-5 | 14.5 | -17.9 | 2384 | UDNA | Mulitza et al. (2008) |
| GeoB9526 | 12.4 | -18.1 | 3223 | UDNA | Zarriess and Mackensen (2011) |
| GIK17049-6 | 55.3 | -26.7 | 3331 | UDNA | Jung and Sarnthein (2003) |
| GIK17051 | 56.2 | -31.9 | 2295 | UDNA | Sarnthein et al. (1994) |
| GIK23415-9 | 53.2 | -19.2 | 2472 | UDNA | Weinelt et al. (2003) |
| KF13 | 37.6 | -31.8 | 2690 | UDNA | Richter (1998) |
| MD95-2040 | 40.6 | -9.9 | 2465 | UDNA | Voelker and de Abreu (2011) |
| MD99-2334 | 37.8 | -10.2 | 3146 | UDNA | Skinner and Shackleton (2004) |
| NA87-22 | 55.5 | -14.7 | 2161 | UDNA | Duplessy et al. (1992) |
| ODP658C | 20.8 | -18.6 | 2274 | UDNA | Woodruff and Chambers (1991) |
| ODP980 | 55.5 | -14.7 | 2168 | UDNA | Oppo et al. (2006) |
| SU90-03 | 40.1 | -32 | 2475 | UDNA | Cortijo et al. (1999) |
| V29-202 | 61 | -21 | 2658 | UDNA | Oppo and Lehman (1995) |
| ENO66-16 | 5.5 | -21.1 | 3152 | UDNA | Oppo and Fairbanks (1987) |
| ENO66-21 | 4.2 | -21.6 | 3995 | UDNA | Oppo and Fairbanks (1987) |
| ENO66-26 | 3.1 | -20 | 4745 | LDNA | Oppo and Fairbanks (1987) |
| ENO66-32 | 2.5 | -19.7 | 4998 | LDNA | Oppo and Fairbanks (1987) |
| ENO66-36 | 4.3 | -20.2 | 4095 | LDNA | Oppo and Fairbanks (1987) |
| ENO66-38 | 4.9 | -20.5 | 2931 | UDNA | Oppo and Fairbanks (1987) |
| ENO66-44 | 5.3 | -21.7 | 3428 | UDNA | Oppo and Fairbanks (1987) |
| GeoB1101 | 1.7 | -10.9 | 4588 | LDNA | Bickert and Wefer (1996) |
| GeoB4216 | 30.6 | -12.4 | 2324 | UDNA | Freudenthal et al. (2002) |
| GIK12328-5 | 21.2 | -18.6 | 2778 | UDNA | Sarnthein et al. (1994) |
| GIK12347-2 | 15.8 | -17.9 | 2576 | UDNA | Sarnthein et al. (1994) |
| GIK12379-3 | 23.1 | -17.8 | 2136 | UDNA | Sarnthein et al. (1994) |
| GIK12392-1 | 25.2 | -16.9 | 2573 | UDNA | Sarnthein et al. (1994) |
| GIK15612-2 | 44.4 | -26.5 | 3050 | UDNA | Sarnthein et al. (1994) |
| GIK15669 | 34.9 | -7.8 | 2022 | UDNA | Sarnthein et al. (1994) |
| GIK15672 | 34.9 | -8.1 | 2460 | UDNA | Sarnthein et al. (1994) |
| GIK16402 | 14.4 | -20.5 | 4202 | LDNA | Sarnthein et al. (1994) |
| GIK17055-1 | 48.2 | -27.1 | 2558 | UDNA | Sarnthein et al. (1994) |
| GIK23414-9 | 53.5 | -20.3 | 2196 | UDNA | Sarnthein et al. (1994) |
| GIK23416-4 | 51.6 | -20 | 3616 | UDNA | Jung and Sarnthein et al. (1994) |
| GIK23417-1 | 50.7 | -19.4 | 3850 | UDNA | Jung and Sarnthein (2004) |




| Core Name | Lat | Lon | Depth (m) | Region | Reference |
|---|---|---|---|---|---|
| GIK23418-8 | 52.6 | -20.3 | 2841 | UDNA | Jung and Sarnthein (2004) |
| IODP-U1308 | 49.9 | -24.2 | 3900 | UDNA | Hodell et al. (2008) |
| KNR110-50 | 4.9 | -43.2 | 3995 | UDNA | Curry et al. (1988) |
| KNR110-55 | 4.9 | -42.9 | 4556 | LDNA | Curry et al. (1988) |
| KNR110-58 | 4.8 | -43 | 4341 | LDNA | Curry et al. (1988) |
| KNR110-66 | 4.6 | -43.4 | 3547 | UDNA | Curry et al. (1988) |
| KNR110-71 | 4.4 | -43.7 | 3164 | UDNA | Curry et al. (1988) |
| KNR110-75 | 4.3 | -43.4 | 3063 | UDNA | Curry et al. (1988) |
| KNR110-82 | 4.3 | -43.5 | 2816 | UDNA | Curry et al. (1988) |
| KNR110-91 | 4.8 | -43.3 | 3810 | UDNA | Curry et al. (1988) |
| MD95-2039 | 40.6 | -10.4 | 3381 | UDNA | Schönfeld et al. (2003) |
| ODP928 | 5.5 | -43.8 | 4012 | LDNA | Curry and Oppo (2005) |
| SU90-39 | 52.5 | -22 | 3955 | UDNA | Chapman and Shackleton (1998) |
| V22-197 | 14.2 | -18.6 | 3167 | UDNA | Boyle (1992) |
| V23-81 | 54.3 | -16.8 | 2393 | UDNA | Veum et al. (1992) |
| V25-59 | 1.4 | -33.5 | 3824 | UDNA | Bertram et al. (1995) |
| V30-49 | 18.4 | -21.1 | 3093 | UDNA | Boyle (1992); Martin and Lea (1998) |
| CH73-139 | 54.7 | -16.4 | 2209 | UDNA | Bickert and Mackensen (2003) |
| CHN82-24 | 43.5 | -30.7 | 3070 | UDNA | Boyle and Keigwin (1985) |
| GIK13521 | 3 | -22 | 4504 | LDNA | Sarnthein et al. (1994) |
| GIK16415 | 9.6 | -19.1 | 3841 | UDNA | Sarnthein et al. (1994) |
| GIK17050 | 55.5 | -27.9 | 2795 | UDNA | Sarnthein et al. (1994) |
| HM52-43 | 63.5 | -0.7 | 2781 | UDNA | Veum et al. (1992) |
| GeoB1041 | -3.5 | -7.6 | 4033 | DSA | Mackensen and Bickert (1999) |
| GeoB1112 | -5.8 | -10.8 | 3125 | DSA | Mackensen and Bickert (1999) |
| GeoB1117 | -3.8 | -14.9 | 3984 | DSA | Mackensen and Bickert (1999) |
| GeoB1118 | -3.6 | -16.4 | 4675 | DSA | Mackensen and Bickert (1999) |
| GeoB1211 | -24.5 | 7.5 | 4089 | DSA | Bickert and Wefer (1999) |
| GeoB1214 | -24.7 | 7.2 | 3210 | DSA | Bickert and Wefer (1999) |
| GeoB1710 | -23.4 | 11.7 | 2987 | DSA | Schmiedl and Mackensen (1997) |
| MD07-3076 | -44.2 | -14.2 | 3770 | DSA | Skinner et al. (2010) |
| ODP1089 | -40.9 | 9.9 | 4621 | DSA | Hodell et al. (2001) |
| ODP1090 | -42.9 | 8.9 | 3702 | DSA | Venz and Hodell (2002) |
| PS2498 | -44.2 | -14.2 | 3783 | DSA | Hodell et al. (2003) |
| RC13-228 | -22.3 | 11.2 | 3204 | DSA | Boyle (1992) |
| V29-135 | -19.7 | 8.88 | 2675 | DSA | Sarnthein et al. (1994) |
| V30-40 | 0.2 | -23.2 | 3706 | DSA | Oppo and Fairbanks (1987) |




| Core Name | Lat | Lon | Depth (m) | Region | Reference |
|---|---|---|---|---|---|
| DSDP502 | 11.5 | -79.4 | 1800 | INA | Demenocal et al. (1992) |
| GeoB6718 | 52.2 | -12.8 | 900 | INA | Rüggeberg et al. (2005) |
| GIK15666-6 | 34.9 | -7.1 | 803 | INA | Sarnthein et al. (1994) |
| GIK16006-1 | 29.3 | -11.5 | 796 | INA | Sarnthein et al. (1994) |
| GIK16017 | 21.3 | -17.8 | 812 | INA | Sarnthein et al. (1994) |
| OCE205-103GGC | 26.1 | -78.1 | 965 | INA | Slowey and Curry (1995) |
| GeoB4240 | 28.9 | -13.2 | 1358 | INA | Freudenthal et al. (2002) |
| GIK11944-2 | 35.6 | -8.1 | 1765 | INA | Weinelt and Sarnthein (2003) |
| GIK16004 | 29.9 | -10.7 | 1512 | INA | Sarnthein et al. (1994) |
| GIK16030 | 21.2 | -18.1 | 1500 | INA | Sarnthein et al. (1994) |
| GIK23419 | 54.9 | -19.8 | 1491 | INA | Sarnthein et al. (1994) |
| GIK23519 | 64.8 | -29.6 | 1893 | INA | Millo et al. (2006) |
| M35003-4 | 12.1 | -61.2 | 1299 | INA | Zahn and Stüber (2002) |
| ODP982 | 57.5 | -15.9 | 1134 | INA | Venz et al. (1999); Venz and Hodell (2002) |
| ODP983 | 60.4 | -23.6 | 1984 | INA | Mc Intyre et al. (1999); Raymo et al. (2004) |
| ODP984 | 61 | -24 | 1650 | INA | Raymo et al. (2004) |
| V28-127 | 11.7 | -80.1 | 1750 | INA | Oppo and Fairbanks (1990) |
| V28-14 | 64.8 | -29.7 | 1855 | INA | Boyle (1992) |
| RC16-84 | -26.7 | -43.3 | 2438 | ISA | Oppo and Horowitz (2000) |
| V24-253 | -26.9 | -44.7 | 2069 | ISA | Oppo and Horowitz (2000) |
| CHN115-70 | -29.9 | -35.6 | 2340 | ISA | Curry and Lohmann (1982) |
| MD96-2080 | -36.3 | 19.5 | 2488 | ISA | Martínez-Méndez et al. (2008) |
| ODP1088 | -41.1 | 13.6 | 2082 | ISA | Hodell et al. (2003) |
| GeoB3104 | -3.7 | -37.7 | 767 | ISA | Arz et al. (1999) |
| BT4 | -4 | 10 | 1000 | ISA | Oppo and Fairbanks (1989) |
| KNR159-36 | -27.5 | -46.5 | 1268 | ISA | Oppo and Horowitz, 2000 |
| RC16-119 | -27.7 | -46.5 | 1567 | ISA | Oppo and Horowitz (2000) |
| GeoB3004 | 14.6 | 52.9 | 1803 | II | Schmiedl and Mackensen (2006) |
| MD01-2378 | -13.1 | 121.8 | 1783 | II | Xu et al. (2008) |
| Orgon4-KS8 | 23.5 | 59.2 | 2900 | DI | Sirocko (1994) |
| SO42-74KL | 14.3 | 57.3 | 3212 | DI | Sirocko et al. (1993) |
| MD97-2151 | 8.7 | 109.9 | 1598 | IP | Wei et al. (2006) |
| FR97-GC12 | -23.6 | 153.8 | 990 | IP | Bostock et al. (2004) |
| V19-27 | -0.5 | -82.1 | 1373 | IP | Mix et al. (1991) |
| EW9504-05 | 32.5 | -118.1 | 1818 | IP | Stott et al. (2000) |
| GIK17961-2 | 8.5 | 112.3 | 1795 | IP | Wang et al. (1999) |
| MD97-2120 | -45.5 | 174.9 | 1210 | IP | Pahnke and Zahn (2005) |





| Core Name | Lat | Lon | Depth (m) | Region | Reference |
| --- | --- | --- | --- | --- | --- |
| NGC102 | 32.3 | 157.9 | 2612 | DP | Ohkushi et al. (2003) |
| ODP807A | 3.6 | 156.6 | 2804 | DP | Zhang et al. (2007) |
| ODP846 | -3.1 | -90.8 | 3296 | DP | Mix et al. (1995); Shackleton et al. (1995) |
| RC13-110 | -0.1 | -95.7 | 3231 | DP | Mix et al. (1991) |
| RC13-114 | -1.7 | -103.6 | 3436 | DP | Marchitto et al. (2005) |
| V24-109 | 0.4 | 158.8 | 2367 | DP | Shackleton et al. (1992) |
| ODP1143 | 9.4 | 113.3 | 2772 | DP | Tian et al. (2002) |





**Table A2.** Correlation coefficients and p-values between records. The upper rows are the raw data, and the bottom rows are the pre-whitened to account for autocorrelated time series. To investigate possible leads/lags between records, we shift the atmospheric $CO_2$ record in 100-year increments relative to the $\delta^{13}C$ stacks and, for brevity, list only the best correlations. All p-values account for reduction in degrees of freedom due to either pre-whitening and/or time shifting.

| Record 1 | Record 2 | $CO_2$ time shift (years) | $r^2$ | |
|---|---|---|---|---|
| $CO_2$ | API $\Delta\delta^{13}C_{I-D}$ | 0 | -0.97 | |
| $CO_2$ | API $\Delta\delta^{13}C_{I-D}$ | +1000 | -0.99 | |
| $CO_2$ | AP $\Delta\delta^{13}C_{I-D}$ | 0 | -0.98 | |
| $CO_2$ | AP $\Delta\delta^{13}C_{I-D}$ | +600 | -0.98 | |
| $CO_2$ | $\Delta\delta^{13}C_{(INA/2)-DP}$ | 0 | -0.98 | |
| $CO_2$ | $\Delta\delta^{13}C_{(INA/2)-DP}$ | $-200$ | -0.98 | |
| $CO_2$ | Global $\delta^{13}C$ stack | 0 | 0.93 | |
| $CO_2$ | Global $\delta^{13}C$ stack | +1000 | 0.95 | |
| Record 1 | Record 2 | $CO_2$ time shift (years) | Pre-whitened $r^2$ | Pre-whitened p-value |
| $CO_2$ | API $\Delta\delta^{13}C_{I-D}$ | 0 | -0.39 | 0.08 |
| $CO_2$ | API $\Delta\delta^{13}C_{I-D}$ | +900 | -0.80 | 0.0003 |
| $CO_2$ | AP $\Delta\delta^{13}C_{I-D}$ | 0 | -0.70 | 0.002 |
| $CO_2$ | AP $\Delta\delta^{13}C_{I-D}$ | +300 | -0.73 | 0.002 |
| $CO_2$ | $\Delta\delta^{13}C_{(INA/2)-DP}$ | 0 | -0.79 | 0.0002 |
| $CO_2$ | $\Delta\delta^{13}C_{(INA/2)-DP}$ | $-200$ | -0.82 | 0.0002 |
| $CO_2$ | Global $\delta^{13}C$ stack | 0 | 0.20 | 0.24 |
| $CO_2$ | Global $\delta^{13}C$ stack | $-500$ | 0.36 | 0.11 |



*Competing interests.* The authors declare that they have no conflict of interest.

*Acknowledgements.* We acknowledge the following colleagues whose advice and input substantially improved drafts of this manuscript: David Lea, Syee Weldeab, Jake Gebbie, Andy Ridgwell, and James Rae. Funding for this work came from NSF grants MGG 0926735 and CDI 1125181.




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
