# Peer review of "Deglacial carbon cycle changes observed in a compilation of 127 benthic $\delta^{13}$ C time series (20-6 ka)"

_Climate of the Past, 2018_

## Referee Comment (RC1) · Anonymous Referee #1 · 11 Apr 2018

The authors have compiled 117 time series of benthic d13C from the world ocean to investigate the deglacial evolution of mean ocean changes in d13C and changes in vertical d13C gradients. The paper provides very interesting insights into the vertical and global deglacial evolution of deep-water d13C. Particularly, for the first time, volume of ocean basins has been taken into account to estimate mean ocean d13C and the mean of specific basins and depth levels. These data products can provide important constraints for model evaluation. The findings are generally convincing, but I have some technical comments regarding the documentation of the methods/data and the correctness of the data/meta data (see points 7/8) in the supplementary files:

1) Age modelling strategy. It is not entirely clear to me, how age models have been created for the individual cores. It seems though, that the age models are all based on a graphic correlation or automated alignment (?) to the stacks provided by Stern and Lisiecki (2014). If so, it is not clear to me why the authors deliberately used a graphic correlation stratigraphy with an error of 1-2 ka, when radiocarbon dates are available, at least for some of the cores. Radiocarbon ages should provide better constraints on the posterior calendar age distributions. It seems that, at least in the tropics, the uncertainties in the reservoir ages are small enough to provide better constraints on age than ages derived from alignment of benthic oxygen isotopes. I recommend that the authors better explain, how they have obtained the age models why they have not (if not) used the individual calendar age distributions of radiocarbon ages that may exist.

2) A few more details should be added to the description of the MC/bootstrapping methods employed to estimate uncertainty. It is not clear, whether age uncertainty has also been considered when producing the stacks (i.e. through time series ensembles), and (if so) what assumption has been made regarding the distribution of age uncertainty (i.e. normally or uniformly distributed). The authors might argue that the age uncertainty is already included through the averaging of several time-uncertain time series to produce the stacks, but on the other hand the number of time series is small for some of the stacks and perhaps not representative for the error.

3) Variations in sea level affect the water depth of the core location by more than 100 m over the considered time period. If core locations are close to the boundary between the intermediate box and the deep box, a core might fall into the shallow box during the LGM and into the deep box at 6 ka. The authors should clearly describe whether water depth always refers to the modern water depth. This is particularly important for figure 2, where apparently LGM and 6 ka symbols have been plotted at the same water depth. Sea level changes should also be taken into account in the animation provided in the supplement.

4) I feel that the data base is not sufficiently documented. Would it be possible to

recreate the stacks just based on the information in the paper? Particularly, I miss information on how the data have been aligned to the stacks. Has this just been done by graphic correlation or has some automatic alignment been involved? If the age models are based on an automatic alignment the methods should be clearly stated. If it is just based on individual age markers and interpolation, these markers should also be included into the data set in the supplement.

5) It should be clearly indicated in the supplement, whether the data are given in "real" depth or in composite depth (e.g. cores GeoB1041-GeoB1214 are probably given in composite depth). While this has very little consequences for the results and implications of this paper, it might cause confusion if other people are using the data later and i.e. use age models/radiocarbon data or other data from the same cores in real depth together with data in composite depth from this study. Also note that these srecords are in many cases actually composed of two cores, a box- or muticore and a gravity core with individual water depths and positions.

6) I also recommend to document not only the references in table A1, but also the actual source (i.e. PANGAEA, NOAA, personal communication) of the data, if possible with doi. This allows to resolve possible inconsistencies later (see 7/8).

7) Is it possible that some of the sites are not labeled correctly in the data files (perhaps I am just reading it wrong)? For example, in the file "DSA_sites.xls", core GeoB1118 (column J) has some Holocene d13C values close to 1 o/oo. I cannot find these values in the data file on Pangaea (https://doi.pangaea.de/10.1594/PANGAEA.103632?format=html#download) or in the original thesis by Bickert (1992, page 187-190) (available under http://elib.suub.uni-bremen.de/ip/docs/00010668.pdf).

8) Also, latitude, longitude and water depth are apparently wrong for some core locations in the corresponding "DSA_xyz.csv" file. For example:

In file DSA_xyz.xls:

GeoB1118: Lat:-3.82, Long:-14.9, WD: 3984 m

In PANGAEA:

GeoB1118-2: Lat: -3.56, Long: -16.431660, WD: -4675.0 m GeoB1118-3: Lat: -3.56, Long: -16.428333, WD: -4671.0 m

Pangaea is correct, I checked with the original cruise report. I have not checked other files, but in DSA_xyz.xls other positions are also wrong (i.e. GeoB1211). Perhaps this is just an unfortunate accidental cell shift, but I suggest to check all data files again, to strictly label all columns of the data files to avoid confusion and to ensure that the analysis of the data has not been affected by the errors in the data.

---

## Referee Comment (RC2) · A. Schmittner (Referee) · 25 Apr 2018

The authors present a compilation of benthic d13C records during the last deglaciation. The work contributes to a better understanding of carbon cycle and ocean circulation changes and is appropriate for Climate of the Past. I think the paper is well written and nicely illustrated. I have a few questions and comments that the authors may want to consider in a revision.

I think typically the relationship between the terrestrial carbon storage and whole-ocean d13C changes is calculated using a closed system approach with land, ocean and atmospheric reservoirs of carbon (e.g. page 1, lines 14-15; Ciais et al. 2012). I

wonder if this is appropriate for glacial-interglacial changes because it is likely that ocean sediments responded by adding/removing alkalinity and carbon from dissolution/accumulation of calcium carbonate. This would also affect d13C of DIC. Is this considered here? It would be good to discuss this point.

Benthic d13C is affected by carbonate ion and pressure effects (e.g. Schmittner et al., 2017). Were these effects considered here? I guess not since carbonate ion changes are not available. In this case it may be useful to try one of their regression equations that don't require carbonate ion to calculate d13C_DIC.

Page 5, 9-11: include study by Schmittner and Somes (2016, Paleoceanography, doi:10.1002/2015PA002905)

Page 7, 12: I didn't find this number (0.15 permil for the standard deviation) in Gebbie et al., (2015). Schmittner et al. (2017) suggest a larger error of ~0.25 permil.

Page 8, 21: "DSA d13C begins increasing at 18 ka" This finding seems to be at odds with Lund et al's (2015, doi:10.1002/2014PA002657) findings that the DSA begins increasing only later (after HS1). Are those data included here? Discuss.

Page 9, 13: The North Pacific (>30N) is also not included.

In Figure 5, which relationship between d13C and land carbon was used? See comment above. Does it consider sediment carbon changes?

Page 10, 9: what volume was used for the deep Pacific box? <30N ?

Page 11, 1-3: Schmittner and Lund (2015, Climate of the Past, doi:10.5194/cp-11-135-2015) have suggested a different mechanism. Please consider.

---

## Author Comment (AC1) · 13 Jun 2018

We thank referee #1 for their helpful comments and especially for bringing technical issues to the author's attention. Our responses (AR) are listed after a synopsis of referee #1's comments (R1).

R1 1) Age modelling strategy. It is not entirely clear to me, how age models have been created for the individual cores. It seems though, that the age models are all based on a graphic correlation or automated alignment (?) to the stacks provided by Stern and Lisiecki (2014). If so, it is not clear to me why the authors deliberately used a graphic correlation stratigraphy with an error of 1-2 ka, when radiocarbon dates are available,

at least for some of the cores. Radiocarbon ages should provide better constraints on the posterior calendar age distributions. It seems that, at least in the tropics, the uncertainties in the reservoir ages are small enough to provide better constraints on age than ages derived from alignment of benthic oxygen isotopes. I recommend that the authors better explain, how they have obtained the age models why they have not (if not) used the individual calendar age distributions of radiocarbon ages that may exist.

AR 1) Thank you for bringing up this important point of discussion. We will clarify these issues during revision of the manuscript text.

Yes, the age models used here are those developed by Stern and Lisiecki (2014, Paleoceanography; hereafter SL14). The SL14 regional age models have 95% confidence intervals of 0.5-2.0 kyr. (A 2-kyr 95% CI width is approximately equivalent to a standard deviation of 0.5 kyr). We did not perform any new $\delta$18O alignments or other age model development for this study.

One of the main goals of this paper is to evaluate whether the SL14 age models are accurate enough to reconstruct the timing of global carbon cycle change. In fact, our results demonstrate that the SL14 age models agree well with ice core $CO_2$ changes which have very well-constrained age models (Monnin et al., 2004; Marcott et al., 2014; Kohler et al., 2017). Research is currently underway to further improve the sediment core age models and their uncertainty estimates, but this work will take 1-2 years and is clearly beyond the scope of the current paper.

Because the reviewer's questions/concerns about our method will likely be shared by many readers, we will add more explanation of how the SL14 age models were created and why we expect them to have similar or better precision than most 14C age models (i.e., except for a handful of low-latitude cores with the most 14C dates). For example, recent work by Khider et al. (2017, Paleoceanography) demonstrates that, for 17 tropical Pacific cores, the uncertainties between radiocarbon ages and $\delta$18O alignments are very similar.

Here is the proposed text to add:

Stern and Lisiecki (2014) created seven regional age models based on all available 14C planktonic dates from each region. Each of the seven regions has an age model based on planktonic 14C measurements from multiple cores; 14C dates are combined across cores by assuming that benthic $\delta$18O is synchronous within each region (but not necessarily between regions). The first step of this process was generating an initial radiocarbon age model for each of 61 cores by using that core's radiocarbon dates, the Bayesian age modeling software Bacon (Blaauw and Christen, 2011), the Marine13 calibration (Reimer et al., 2013), and constant 405 14C-yr reservoir ages. Bacon was used to estimate 14C-based ages at specified depths throughout each core, including Monte Carlo uncertainty estimates that increase with distance from the 14C measurements. To identify the core-specific depths for which 14C-based ages would be combined, each core's benthic $\delta$18O record was aligned to an Atlantic or Pacific target core using the alignment software Match (Lisiecki and Lisiecki, 2002). Creating regional age models maximizes the total number of 14C dates which contribute to each age model. For example, the intermediate Pacific age model is derived from 14 sediment cores that include a total of 160 radiocarbon dates. The $\delta$13C records analyzed here use the age models produced by Stern & Lisiecki (2014), which were created by converting each core's Match-based $\delta$18O alignment to its region's radiocarbon age model.

Additionally, Stern and Lisiecki (2014) estimate 95% confidence intervals for each regional age model using 10,000 Monte Carlo age samples for each core from Bacon. Age uncertainty estimates for each region include the effects of any errors in benthic $\delta$18O alignment because alignment errors would increase scatter in the compiled radiocarbon dates (by aligning portions of cores with different ages) and, thus, increase the observed spread in age estimates. For the time range of 0-20 kyr used in our $\delta$13C compilation, the 95% confidence interval widths of the regional age models range from 0.5-2.0 kyr. Although Match does not quantify alignment uncertainty, alignment uncertainties have been estimated using a similar algorithm, called HMM-Match (Lin et

al., 2014). For age models generated either by $\delta$18O -alignments or radiocarbon, the amount of age uncertainty depends on the time resolution of the $\delta$18O or 14C data, respectively. A comparison of 15 low-latitude Pacific cores found that 14C-based age uncertainty is comparable to, if not greater than, the uncertainty associated with $\delta$18O alignments by HMM-Match (Khider et al., 2017, Paleoceanography).

R1 2) ... It is not clear, whether age uncertainty has also been considered when producing the stacks (i.e. through time series ensembles), and (if so) what assumption has been made regarding the distribution of age uncertainty (i.e. normally or uniformly distributed). The authors might argue that the age uncertainty is already included through the averaging of several time-uncertain time series to produce the stacks, but on the other hand the number of time series is small for some of the stacks and perhaps not representative for the error.

AR 2) The contribution of age uncertainty is important to consider. In the manuscript text, we intend to clarify the descriptions of our methods and the sources of uncertainty, including our bootstrapped Monte Carlo uncertainty estimates. Specifically, estimates of 0.1-0.25‰ in foraminiferal $\delta$13C estimates implicitly include the effects of age model uncertainty. For example, Marchal and Curry (2008) report a standard deviation of 0.1‰ which "includes errors in sediment core chronology and oceanic representativity of benthic $\delta$13C, which alone appears better than this value on average". Additional discussion of the most appropriate $\delta$13C uncertainty is presented in response to Q7 of reviewer 2. We have decided to increase the $\delta$13C uncertainty to 0.2‰ which will slightly increase our reported uncertainties.

Additionally, we address age model uncertainty by calculating the correlations between our $\delta$13C results and ice core CO2 for a range of time lags. Because age model errors would be expected to weaken the correlation between these two archives, the comparison we present is conservative, and improved age models would likely strengthen the observed correlations. During revision, we will add more discussion throughout the manuscript of the potential impact of age uncertainty.

R1 3) Variations in sea level affect the water depth of the core location by more than 100 m over the considered time period. If core locations are close to the boundary between the intermediate box and the deep box, a core might fall into the shallow box during the LGM and into the deep box at 6 ka. The authors should clearly describe whether water depth always refers to the modern water depth. This is particularly important for figure 2, where apparently LGM and 6 ka symbols have been plotted at the same water depth. Sea level changes should also be taken into account in the animation provided in the supplement.

AR 3) This is a good point to bring up that will be clarified in the manuscript text. Sea level changes are not included in our volume weighting – modern sea level and volumes (calculations based on GEBCO 2014, https://doi.org/10.1002/2015EA000107) are used throughout the deglaciation because incorporating sea level changes in our calculations would likely introduce undesirable data artifacts in the regional stacks. The depth-boundary between intermediate and deep boxes would need to shift every 1 kyr, and some sites near the boundary would move from an intermediate to deep box across the deglaciation. Because core coverage is relatively sparse, only a few cores would be affected. If we allowed cores to jump between regions as sea level rises, we would alter the spatial representation of cores in the regions, potentially creating artificial jumps in the regional $\delta13C$ stack when these cores switched regions.

R1 4) Data base documentation: how have the data been aligned to the stacks. Has this just been done by graphic correlation or has some automatic alignment been involved? If the age models are based on an automatic alignment the methods should be clearly stated. If it is just based on individual age markers and interpolation, these markers should also be included into the data set in the supplement.

AR 4) As described in the response to question 1, the age models used here are those developed by Stern and Lisiecki (2014, Paleoceanography; hereafter SL14). We will revise the manuscript to clarify that we did not perform any new $\delta18O$ alignments or other age model development for this study. Therefore, aside from a summary of

the methods of SL14, no additional age model documentation should be required. The supplemental materials include both core depth and estimated age for each $\delta$13C measurement in each core.

R1 5) Real depth vs. composite depth?

AR 5) Details such as composite depth vs. real depth are not always clearly documented when data are downloaded from a repository, but we have done our best to preserve the integrity of the original data. We plan to include a doi identifier or URL for each core's record so that readers will have the ability to find the original documentation for each $\delta$13C record used (see next question/reply). Additionally, the distinction between real depth versus composite depth is usually quite small in the top 20-kyr of the record. Regardless of whether real or composite depth is used, our age estimates will be accurate as long as we use the same depth scale that SL14 used when producing their age models. Another indication that the depth scale for each core has been processed in a consistent manner is the observed similarity of $\delta$13C records among all cores in each region.

R1 6) I also recommend to document not only the references in table A1, but also the actual source (i.e., PANGAEA, NOAA, personal communication) of the data, if possible with doi. This allows to resolve possible inconsistencies later (see 7/8).

AR 6) Although the reviewer brings up a good point, and outlines what will hopefully become part of the best-practices in our field, we did not keep track of the download location or doi for every site. However, we will try to gather this information for as many of the sites as possible within the time constraints of the manuscript revision. When we finish gathering this meta-data, we will add it to the data compilation that will be uploaded to NOAA and Pangaea.

R1 7-8) Site labels, location info.

AR 7-8) We thank the reviewer for finding inconsistencies in the supplemental files. It

is challenging to find every mistake in such a large database without concerted effort. Before submitting the revised manuscript, we will track down and correct any errors in the coordinate information or records from the core sites in this compilation.

—————————————————————

---

## Author Comment (AC2) · 13 Jun 2018

We thank referee #2, Andreas Schmittner for his comments and improvements to the manuscript. Our responses (AR) are listed after a synopsis of Andreas' comments (R2).

R2 1) I think typically the relationship between the terrestrial carbon storage and whole-ocean $\delta$13C changes is calculated using a closed system approach with land, ocean and atmospheric reservoirs of carbon (e.g. page 1, lines 14-15; Ciais et al. 2012). I wonder if this is appropriate for glacial-interglacial changes because it is likely that ocean sediments responded by adding/removing alkalinity and carbon from dissolution/accumulation of calcium carbonate. This would also affect $\delta$13C of DIC. Is this considered here? It would be good to discuss this point.

In Figure 5, which relationship between $\delta$13C and land carbon was used? See comment above. Does it consider sediment carbon changes?

AR 1) Our "global mean $\delta$13C" is a volume-weighted average of benthic $\delta$13C (per mil). In the discussion, we refrain from converting the $\delta$13C estimate to terrestrial carbon storage because it is difficult to propagate errors through the mass balance calculations. To avoid making this conversion, we use two separate y-axes when comparing terrestrial carbon storage change and benthic $\delta$13C change in Figure 5. We will modify the text to clarify that the magnitude of carbon storage and benthic $\delta$13C change is not necessarily equivalent, and that Figure 5 is not meant to be a quantitative comparison. In this manuscript, we are simply noting the remarkable similarity in the pattern of change between our data compilation and the only model results of terrestrial carbon storage change across the deglaciation that we were able to find before submission.

R2 2) Benthic $\delta$13C is affected by carbonate ion and pressure effects (e.g. Schmittner et al., 2017). Were these effects considered here? I guess not since carbonate ion changes are not available. In this case it may be useful to try one of their regression equations that don't require carbonate ion to calculate d13C_DIC

AR 2) That is correct, the effects of carbonate ion are not considered here because estimates of past carbonate ion concentration are currently difficult to constrain. We will make this clear in the revised text and cite the regressions from Schmittner et al. (2017) as way to account for carbonate ion changes if carbonate ion data becomes available or if readers want to consider model-based estimates of carbonate ion concentration. We will also mention the available regressions that can be used when carbonate ion conc. records are unavailable. Because the relevant experiments (LW6 and CW6) suggest a linear scaling of $\delta$13C (i.e., do not include a depth-dependent term), applying these regressions would not impact the correlation coefficients in our manuscript.

Additionally, we will mention that these regressions would impact the apparent scaling between $\delta$13C and terrestrial carbon implied by Figure 5. We prefer not to apply the regression scaling to $\delta$13C in this figure because it could confuse readers and cause misinterpretation of our results.

R2 3) Page 8, 21: "DSA $\delta$13C begins increasing at 18 ka" This finding seems to be at odds with Lund et al's (2015, doi:10.1002/2014PA002657) findings that the DSA begins increasing only later (after HS1). Are those data included here? Discuss.

AR 3) The records from Lund et al. (2015) were not originally included in the manuscript because they were published after Stern & Lisiecki (2014) created their original compilation and were not included in their regional stack age models. We have now included 12 of the Brazil Margin sites from Lund et al. (2015) using their 14C age models. This doesn't change our overall results or the timing of $\delta$13C changes in the DSA region. The DSA $\delta$13C value at the LGM is slightly more depleted than the original version without these sites, but the timing of deglacial increase is visually similar to before. We didn't do any formal change point analysis to quantify the timing of $\delta$13C changes because the focus of this paper is on comparing the global $\delta$13C gradient with the ice core $CO_2$ record throughout the entire deglacial transition.

R2 4) Page 9, 13: The North Pacific (>30N) is also not included. Page 10, 9: what volume was used for the deep Pacific box? <30N?

AR 4) We have two Pacific records at $\sim$32N (one each in the intermediate and deep Pacific regions), but none further north. However, our Pacific volume estimates are based on the latitude range of 60S to 60N. Therefore, the North Pacific (>30N) is volumetrically included but not well constrained.

We are aware of efforts currently underway to compile and publish North Pacific $\delta$13C records, and we expect these data will slightly alter mean global $\delta$13C estimates. However, our compilation and its comparison to ice core $CO_2$ provides an important scientific contribution in its current form, and our analyses can be revisited as additional

data become available.

R2 5) Page 11, 1-3: Schmittner and Lund (2015, Climate of the Past, doi:10.5194/cp-11-135- 2015) have suggested a different mechanism. Please consider.

AR 5) We have done our best to include a variety of hypotheses, but the number of possible citations is quite large. On Page 11, 1-3, we will revise the text to include the hypothesis that AMOC shutdown induced a decline of biologically sequestered ocean carbon storage (Schmittner and Lund, 2015).

R2 6) Page 5, 9-11: include study by Schmittner and Somes (2016, Paleoceanography, doi:10.1002/2015PA002905)

AR 6) Thank you for the suggestion, we will include the citation.

R2 7) Page 7, 12: I didn't find this number (0.15‰ for the standard deviation) in Gebbie et al., (2015). Schmittner et al. (2017) suggest a larger error of ∼0.25‰

AR 7) The error estimate of 0.15‰ we used is actually a compromise between the errors reported by Gebbie et al. (2015)'s 0.20‰ and Marchal and Curry (2008)'s 0.10‰.However, during revision we plan to change our uncertainty estimate to 0.20‰ which will increase our stack 95% confidence intervals by +/- 0.02‰.An uncertainty of 0.20‰ will also be approximately consistent with Schmittner et al. (2017); we will add this citation and an explanation of why the value of 0.20‰ was selected. Specifically, the most relevant uncertainty from Schmittner et al. (2017) would likely be a standard deviation 0.22‰ as observed in experiments LW6 and CW6 in Table 2 of that paper because those experiments are the ones that include only C. wuellerstorfi. To the extent that modern-day observations also contribute uncertainty in comparison with late Holocene $\delta$13C, 0.20‰ is a reasonable estimate of the uncertainty contribution from foraminiferal $\delta$13C. Additionally, the largest discrepancies between foram $\delta$13C and DIC $\delta$13C in Schmittner's compilation (their Figure 4) comes from shallow cores (<1 km) and high-latitude regions (especially the Arctic), which are not included in our

compilation.

---

## Author Comment (AC3) · 13 Jun 2018

Dear Carlye and Lorraine,

I have missed the discussion phase of your CPD paper

Peterson, C. and Lisiecki, L.: Deglacial carbon cycle changes observed in a compilation of 117 benthic $\delta$13C time series (20-6 ka), Clim. Past Discuss., https://doi.org/10.5194/cp-2018-25, in review, 2018.

Therefore, I like to give you 1 comment via email:

When comparing Dd13C with CO2 (your figs 7, A1c) you might consider the CO2

stack I compiled last year, since I tried to find an objective way to deal with the offsets between the different ice cores and used the most recent age models. Maybe you might also prefer our calculated spline for your comparison.

Köhler, P., Nehrbass-Ahles, C., Schmitt, J., Stocker, T. F., and Fischer, H.: A 156 kyr smoothed history of the atmospheric greenhouse gases CO2, CH4, and N2O and their radiative forcing, Earth Syst. Sci. Data, 9, 363-387, https://doi.org/10.5194/essd-9-363-2017, 2017. (link to data in the abstract).

Looking forward for your final paper. Best Peter

– Dr. Peter Köhler Alfred Wegener Institute Helmholtz Centre for Polar and Marine Research PO Box 12 01 61, D-27515 Bremerhaven, Germany Phone: +49 471 4831 1687 Fax: +49 471 4831 1149 Email: Peter.Koehler@awi.de http://www.awi.de/People/show?pkoehler

Our response: We thank Peter Köhler for his suggestion to use the CO2 compilation from his 2017 paper to compare against our $\delta$13C stacks. We have compared the $\delta$13C stacks to the Köhler 2017 CO2 compilation, and it produces very similar correlation coefficients as the spliced CO2 record used in our original draft. Therefore, we plan to revise the manuscript using Kohler's CO2 compilation.
* * *

---

## Author Response (AR1)

List of major changes to manuscript:
- Updated all figures, captions, statistics, and text to reflect comparison to Kohler (2017) atmospheric CO2 records.
- Revised the manuscript as suggested by Reviewer #1 and Andreas Schmittner. Details below, revised sections marked in **bold**.
- Added 10 Lund et al (2014) South Atlantic sites to compilation and throughout manuscript. Compilation is now at 127 sites.
- As suggested by Reviewer #1, we added a "DOI" column to table A1 and a supplemental file called "TableA1_DOI.xlsx and .csv" so readers can find the original source for the records in this compilation. We are continuing to work on formatting this d13C compilation for archival on both NOAA and Pangaea.
- Table A1 was altered by: adding data DOI/URLs, updating references, rearranging records into regions and descending depth within a region, and double checked spatial coordinates for DSA sites (spot checked other sites). Within the manuscript, table A1 has a very small font size to fit the references and DOI/URLs. Hopefully the Copernicus editorial staff can help in later stages.
- The "tracked changes" style comparison is at the end of this document, but the references compiled in a strange way, so the newly added references show up as "?". In the comments to reviewer responses, we note the section where new citations are added.
* * *
Below are details of how we addressed reviewer suggestions, including previous rebuttal and new comments with revised sections marked in **bold**.

We thank referee #1 for their helpful comments and especially for bringing technical issues to the author's attention. Our responses (AC) are listed after a synopsis of referee #1's comments.

R1 1) Age modelling strategy. It is not entirely clear to me, how age models have been created for the individual cores. It seems though, that the age models are all based on a graphic correlation or automated alignment (?) to the stacks provided by Stern and Lisiecki (2014). If so, it is not clear to me why the authors deliberately used a graphic correlation stratigraphy with an error of 1-2 ka, when radiocarbon dates are available, at least for some of the cores. Radiocarbon ages should provide better constraints on the posterior calendar age distributions. It seems that, at least in the tropics, the uncertainties in the reservoir ages are small enough to provide better constraints on age than ages derived from alignment of benthic oxygen isotopes. I recommend that the authors better explain, how they have obtained the age models why they have not (if not) used the individual calendar age distributions of radiocarbon ages that may exist.

**AC 1)** Thank you for bringing up this important point of discussion. We have clarified these issues during revision of the manuscript text **in Section 4.2 "Stacking"**.

Yes, the age models used here are those developed by Stern and Lisiecki (2014, Paleoceanography; hereafter SL14). The SL14 regional age models have 95% confidence intervals of 0.5-2.0 kyr. (A 2-kyr 95% CI width is approximately equivalent to a standard deviation of 0.5 kyr). We did not perform any new $\delta^{18}O$ alignments or other age model development for this study.

One of the main goals of this paper is to evaluate whether the SL14 age models are accurate enough to reconstruct the timing of global carbon cycle change. In fact, our results demonstrate that the SL14 age models agree well with ice core $CO_2$ changes which have very well-constrained age models (Monnin et al., 2004; Marcott et al., 2014; Kohler et al., 2017). Research is currently underway to further improve the sediment core age models and their uncertainty estimates, but this work will take 1-2 years and is clearly beyond the scope of the current paper.

Because the reviewer's questions/concerns about our method will likely be shared by many readers, we will add more explanation of how the SL14 age models were created and why we expect them to have similar or better precision than most $^{14}C$ age models (i.e., except for a handful of low-latitude cores with the most $^{14}C$ dates). For example, recent work by Khider et al. (2017, Paleoceanography) demonstrates that, for 17 tropical Pacific cores, the uncertainties between radiocarbon ages and $\delta^{18}O$ alignments are very similar.

Here is the proposed text to add:
Stern and Lisiecki (2014) created seven regional age models based on all available $^{14}C$ planktonic dates from each region. Each of the seven regions has an age model based on planktonic $^{14}C$ measurements from multiple cores; $^{14}C$ dates are combined across cores by assuming that benthic $\delta^{18}O$ is synchronous within each region (but not necessarily between regions). The first step of this process was generating an initial radiocarbon age model for each of 61 cores by using that core's radiocarbon dates, the Bayesian age modeling software Bacon (Blaauw and Christen, 2011), the Marine13 calibration (Reimer et al., 2013), and constant 405 $^{14}C$-yr reservoir ages. Bacon was used to estimate $^{14}C$-based ages at specified depths throughout each core, including Monte Carlo uncertainty estimates that increase with distance from the $^{14}C$ measurements. To identify the core-specific depths for which $^{14}C$-based ages would be combined, each core's benthic $\delta^{18}O$ record was aligned to an Atlantic or Pacific target core using the alignment software Match (Lisiecki and Lisiecki, 2002). Creating regional age models maximizes the total number of $^{14}C$ dates which contribute to each age model. For example, the intermediate Pacific age model is derived from 14 sediment cores that include a total of 160 radiocarbon dates. The $\delta^{13}C$ records analyzed here use the age models produced by Stern & Lisiecki (2014), which were created by converting each core's Match-based $\delta^{18}O$ alignment to its region's radiocarbon age model.

Additionally, Stern and Lisiecki (2014) estimate 95% confidence intervals for each regional age model using 10,000 Monte Carlo age samples for each core from Bacon. Age uncertainty estimates for each region include the effects of any errors in benthic $\delta^{18}O$ alignment because alignment errors would increase scatter in the compiled radiocarbon dates (by aligning portions

of cores with different ages) and, thus, increase the observed spread in age estimates. For the time range of 0-20 kyr used in our $\delta^{13}$C compilation, the 95% confidence interval widths of the regional age models range from 0.5-2.0 kyr. Although Match does not quantify alignment uncertainty, alignment uncertainties have been estimated using a similar algorithm, called HMM-Match (Lin et al., 2014). For age models generated either by $\delta^{18}$O -alignments or radiocarbon, the amount of age uncertainty depends on the time resolution of the $\delta^{18}$O or $^{14}$C data, respectively. A comparison of 15 low-latitude Pacific cores found that $^{14}$C-based age uncertainty is comparable to, if not greater than, the uncertainty associated with $\delta^{18}$O alignments by HMM-Match (Khider et al., 2017, Paleoceanography).

R1 2) ... It is not clear, whether age uncertainty has also
been considered when producing the stacks (i.e. through time series ensembles), and
(if so) what assumption has been made regarding the distribution of age uncertainty
(i.e. normally or uniformly distributed). The authors might argue that the age uncertainty
is already included through the averaging of several time-uncertain time series to
produce the stacks, but on the other hand the number of time series is small for some
of the stacks and perhaps not representative for the error.

**AC 2)** The contribution of age uncertainty is important to consider. For clarification, **we added a new section 4.3 "Stack limitations and uncertainty".** Additionally, **in section 4 "Methods"**, we clarified the descriptions of our methods and the sources of uncertainty, including our bootstrapped Monte Carlo uncertainty estimates. Specifically, estimates of 0.1-0.25‰ in foraminiferal $\delta^{13}$C estimates implicitly include the effects of age model uncertainty. For example, Marchal and Curry (2008) report a standard deviation of 0.1‰ which "includes errors in sediment core chronology and oceanic representativity of benthic $\delta^{13}$C, which alone appears better than this value on average". Additional discussion of the most appropriate $\delta^{13}$C uncertainty is presented in response to Q7 of reviewer 2. We have decided to increase the $\delta^{13}$C uncertainty to 0.2‰, which will slightly increase our reported uncertainties.

Additionally, we address age model uncertainty by calculating the correlations between our $\delta^{13}$C results and ice core $CO_2$ for a range of time lags. Because age model errors would be expected to weaken the correlation between these two archives, the comparison we present is conservative, and improved age models would likely strengthen the observed correlations. During revision, we will add more discussion throughout the manuscript of the potential impact of age uncertainty.

R1 3) Variations in sea level affect the water depth of the core location by more than 100 m over the considered time period. If core locations are close to the boundary between the intermediate box and the deep box, a core might fall into the shallow box during the LGM and into the deep box at 6 ka. The authors should clearly describe whether water depth always refers to the modern water depth. This is particularly important for figure 2, where apparently LGM and 6 ka symbols have been plotted at the same water depth. Sea level changes should also be taken into account in the animation provided

in the supplement.

**AC 3)** This is a good point to bring up that will be clarified in the manuscript text **in section 4.2 "Stacking".** Sea level changes are not included in our volume weighting -- modern sea level and volumes (calculations based on GEBCO 2014, **https://doi.org/10.1002/2015EA000107**) are used throughout the deglaciation because incorporating sea level changes in our calculations would likely introduce undesirable data artifacts in the regional stacks. The depth-boundary between intermediate and deep boxes would need to shift every 1 kyr, and some sites near the boundary would move from an intermediate to deep box across the deglaciation. Because core coverage is relatively sparse, only a few cores would be affected. If we allowed cores to jump between regions as sea level rises, we would alter the spatial representation of cores in the regions, potentially creating artificial jumps in the regional $\delta^{13}C$ stack when these cores switched regions.

R1 4) Data base documentation: how have the data been aligned to the stacks. Has this just been done by graphic correlation or has some automatic alignment been involved? If the age models are based on an automatic alignment the methods should be clearly stated. If it is just based on individual age markers and interpolation, these markers should also be included into the data set in the supplement.

**AC 4)** As described in the response to question 1, the age models used here are those developed by Stern and Lisiecki (2014, Paleoceanography; hereafter SL14). **Section 4.1** clarifies that we did not perform any new $\delta^{18}O$ alignments or other age model development for this study. Therefore, aside from a summary of the methods of SL14, no additional age model documentation should be required. The supplemental materials include both core depth and estimated age for each $\delta^{13}C$ measurement in each core.

R1 5) Real depth vs. composite depth?

**AC 5)** Details such as composite depth vs. real depth are not always clearly documented when data are downloaded from a repository, but we have done our best to preserve the integrity of the original data. **We document the DOI identifier or URL for each core's record in Table A1 and in the folder for additional supplemental files "TableA1_DOI.xlsx and .csv"** so that readers will have the ability to find the original documentation for each $\delta^{13}C$ record used (see next question/reply). Additionally, the distinction between real depth versus composite depth is usually quite small in the top 20-kyr of the record. Regardless of whether real or composite depth is used, our age estimates will be accurate as long as we use the same depth scale that SL14 used when producing their age models. Another indication that the depth scale for each core has been processed in a consistent manner is the observed similarity of $\delta^{13}C$ records among all cores in each region.

R1 6) I also recommend to document not only the references in table A1, but also the actual source (i.e., PANGAEA, NOAA, personal communication) of the data, if possible with doi. This allows to resolve possible inconsistencies later (see 7/8).

**AC 6)** We **document the DOI or URL for the original data for the records in supplemental table A1 and in the supplemental folder "TableA1_DOI".**

Although the reviewer brings up a good point, and outlines what will hopefully become part of the best-practices in our field, we did not keep track of the download location or doi for every site. However, we will try to gather this information for as many of the sites as possible within the time constraints of the manuscript revision. When we finish gathering this meta-data, we will add it to the data compilation that will be uploaded to NOAA and Pangaea.

R1 7-8) Site labels, location info.

**AC 7-8) We were able to correct these mistakes.** We thank the reviewer for finding inconsistencies in the supplemental files. It is challenging to find every mistake in such a large database without concerted effort. Before submitting the revised manuscript, we will track down and correct any errors in the coordinate information or records from the core sites in this compilation.
* * *
We thank referee #2, **Andreas Schmittner** for his comments and improvements to the manuscript. Our responses (AC) are listed after a synopsis of Andreas' comments (R2).

R2 1) I think typically the relationship between the terrestrial carbon storage and whole-ocean $\delta^{13}C$ changes is calculated using a closed system approach with land, ocean and atmospheric reservoirs of carbon (e.g. page 1, lines 14-15; Ciais et al. 2012). I wonder if this is appropriate for glacial-interglacial changes because it is likely that ocean sediments responded by adding/removing alkalinity and carbon from dissolution/accumulation of calcium carbonate. This would also affect $\delta^{13}C$ of DIC. Is this considered here? It would be good to discuss this point.

In Figure 5, which relationship between $\delta^{13}C$ and land carbon was used? See comment above. Does it consider sediment carbon changes?

**AC 1) We clarify these points in section 5.1 and figure 5.**

Our "global mean $\delta^{13}C$" is a volume-weighted average of benthic $\delta^{13}C$ (per mil). In the discussion, we refrain from converting the $\delta^{13}C$ estimate to terrestrial carbon storage because it is difficult to propagate errors through the mass balance calculations. To avoid making this conversion, we use two separate y-axes when comparing terrestrial carbon storage change and benthic $\delta^{13}C$ change in Figure 5. We will modify the text to clarify that the magnitude of carbon storage and benthic $\delta^{13}C$ change is not necessarily equivalent, and that Figure 5 is not meant to be a quantitative comparison. In this manuscript, we are simply noting the remarkable similarity in the pattern of change between our data compilation and the only model results of

terrestrial carbon storage change across the deglaciation that we were able to find before submission.

R2 2) Benthic $\delta^{13}$C is affected by carbonate ion and pressure effects (e.g. Schmittner et al., 2017). Were these effects considered here? I guess not since carbonate ion changes are not available. In this case it may be useful to try one of their regression equations that don't require carbonate ion to calculate d13C_DIC

**AC 2) We address this is section 4.3.**

That is correct, the effects of carbonate ion are not considered here because estimates of past carbonate ion concentration are currently difficult to constrain. We will make this clear in the revised text and cite the regressions from Schmittner et al. (2017) as way to account for carbonate ion changes if carbonate ion data becomes available or if readers want to consider model-based estimates of carbonate ion concentration. We will also mention the available regressions that can be used when carbonate ion conc. records are unavailable. Because the relevant experiments (LW6 and CW6) suggest a linear scaling of $\delta^{13}$C (i.e., do not include a depth-dependent term), applying these regressions would not impact the correlation coefficients in our manuscript. Additionally, we will mention that these regressions would impact the apparent scaling between $\delta^{13}$C and terrestrial carbon implied by Figure 5. We prefer not to apply the regression scaling to $\delta^{13}$C in this figure because it could confuse readers and cause misinterpretation of our results.

R2 3) Page 8, 21: "DSA $\delta^{13}$C begins increasing at 18 ka" This finding seems to be at odds with Lund et al's (2015, doi:10.1002/2014PA002657) findings that the DSA begins increasing only later (after HS1). Are those data included here? Discuss.

**AC 3)** We **added 10 sites from Lund et al. 2015** that were not originally included in the manuscript and used their $^{14}$C age models.

The records from Lund et al. (2015) were not originally included in the manuscript because they were published after Stern & Lisiecki (2014) created their original compilation and were not included in their regional stack age models. We have now included 12 of the Brazil Margin sites from Lund et al. (2015) using their $^{14}$C age models. This doesn't change our overall results or the timing of $\delta^{13}$C changes in the DSA region. The DSA $\delta^{13}$C value at the LGM is slightly more depleted than the original version without these sites, but the timing of deglacial increase is visually similar to before. We didn't do any formal change point analysis to quantify the timing of $\delta^{13}$C changes because the focus of this paper is on comparing the global $\delta^{13}$C gradient with the ice core $CO_2$ record throughout the entire deglacial transition.

R2 4) Page 9, 13: The North Pacific (>30N) is also not included. Page 10, 9: what volume was used for the deep Pacific box? <30N?

**AC 4)** We have two Pacific records at ~32N (one each in the intermediate and deep Pacific regions), but none further north. However, our Pacific volume estimates are based on the latitude range of 60S to 60N. Therefore, the North Pacific (>30N) is volumetrically included but not well constrained.

We are aware of efforts currently underway to compile and publish North Pacific $\delta^{13}C$ records, and we expect these data will slightly alter mean global $\delta^{13}C$ estimates. However, our compilation and its comparison to ice core $CO_2$ provides an important scientific contribution in its current form, and our analyses can be revisited as additional data become available.

R2 5) Page 11, 1-3: Schmittner and Lund (2015, Climate of the Past, doi:10.5194/cp-11-135-2015) have suggested a different mechanism. Please consider.

**AC 5) Citation added to section 6.3.**
We have done our best to include a variety of hypotheses, but the number of possible citations is quite large. On Page 11, 1-3, we will revise the text to include the hypothesis that AMOC shutdown induced a decline of biologically sequestered ocean carbon storage (Schmittner and Lund, 2015).

R2 6) Page 5, 9-11: include study by Schmittner and Somes (2016, Paleoceanography, doi:10.1002/2015PA002905)

**AC 6) Citation added to section 2.3.**
Thank you for the suggestion, we will include the citation.

R2 7) Page 7, 12: I didn't find this number (0.15‰ for the standard deviation) in Gebbie et al., (2015). Schmittner et al. (2017) suggest a larger error of ~0.25‰.

**AC 7) This is clarified in section 4.3.**

The error estimate of 0.15‰ we used is actually a compromise between the errors reported by Gebbie et al. (2015)'s 0.20‰ and Marchal and Curry (2008)'s 0.10‰. However, during revision we plan to change our uncertainty estimate to 0.20‰, which will increase our stack 95% confidence intervals by +/- 0.02‰. An uncertainty of 0.20‰ will also be approximately consistent with Schmittner et al. (2017); we will add this citation and an explanation of why the value of 0.20‰ was selected. Specifically, the most relevant uncertainty from Schmittner et al. (2017) would likely be a standard deviation 0.22‰ as observed in experiments LW6 and CW6 in Table 2 of that paper because those experiments are the ones that include only *C. wuellerstorfi*. To the extent that modern-day observations also contribute uncertainty in comparison with late Holocene $\delta^{13}C$, 0.20‰ is a reasonable estimate of the uncertainty contribution from foraminiferal $\delta^{13}C$. Additionally, the largest discrepancies between foram $\delta^{13}C$ and DIC $\delta^{13}C$ in Schmittner's compilation (their Figure 4) comes from shallow cores (<1 km) and high-latitude regions (especially the Arctic), which are not included in our compilation.

[revised manuscript text omitted]

---

## Author Response (AR2)

Dear Lukas,

Thank you for all your comments which have improved this manuscript and thank you for accepting it. Below are comments or notes on how we have changed the manuscript. We would prefer the key figure to be the deglacial animation in the supplemental files.

-Carlye and Lorraine

**Editor Decision: Publish subject to technical corrections** (16 Jul 2018) by Lukas Jonkers
Comments to the Author:
Dear Carlye,

I have read the revised version of your manuscript on deglacial carbon cycle changes. You have responded well to the reviewers and I am happy to accept the manuscript after a few minor changes.

I appreciate your effort to compile the source information for all the records and thank you again for submitting your paper to our special issue.

kind regards, Lukas Jonkers

Suggested minor changes:
P1, L2: change 'and well-suited' to improve flow of sentence.

   Ok

P5, L18: insert 'the' between 'to' and 'deglacoal', italicise p in pCO2

   Ok

P5, L19: shouldn't it be François et al 1997, instead of Franois?

   Yes

P6, L23: I assume that the other time series from the South Atlantic did not meet the criteria. Is that correct? Can you add a short sentence why only these were added and no other records published since 2014?

You are correct. For example, one South Atlantic site was excluded because it was shallower than 500 m. Also, there are two sites missing one d13C data point after interpolation: Site KNR159-5-22GGC in the DSA at 6 ka, and Site KNR159-5-90GGC in the ISA at 20 ka.

**Added** to Sec. 4.1: "The bulk of data compilation work for this study occurred in 2010-2015, and more recently published data are not included."

**Added** to Sec. 4.2: "The $\delta^{13}$C time series for sites KNR159-5-90GGC and KNR159-5-22GGC do not include 20 ka and 6 ka, respectively \citep{lund2015southwest}; at these times, the relevant sites were excluded from the regional average (Figure 2 and supplemental animation)."

P6, L32: add 'dating' after 'radiocarbon'.

   Ok

P8, L24: please add a brief description of the rationale behind using this metric.

**Previous text:** Additionally, we construct an alternate d13C gradient based on the difference between half the intermediate North Atlantic stack and the deep Pacific stack (Dd13C(INA/2)-DP ), analogous to the gradient compared to CO2 in Lisiecki (2010).

**Changed to:** Additionally, we evaluate an alternate gradient, $\Delta\delta^{13}C_{(INA/2)-DP}$, defined as the difference between half the intermediate North Atlantic stack and the deep Pacific stack; \citep{lisiecki2010d13c} found that this gradient optimized correlation with CO$_2$ from 0-800 ka.

Table 1: add description of what AP stands for
Table 1: I don't seem to be able to find a subscript m in the table

We **changed** subscript 'm' to subscript 'AP' for more consistent nomenclature throughout, and now have updated the footnote description to clarify that all numbers in that 'AP' row do not include Indian Ocean regions.